

# Harmonizing plant functional type distributions for evaluating Earth System Models

Anne Dallmeyer[1], Martin Claussen[1,2] and Victor Brovkin[1]

[1]Max Planck Institute for Meteorology, Bundesstrasse 53, 20146 Hamburg, Germany
[2]Meteorological Institute, Centrum für Erdsystemforschung und Nachhaltigkeit (CEN), Universität Hamburg, Bundesstrasse 55, 20146 Hamburg, Germany

*Correspondence to*: Anne Dallmeyer (anne.dallmeyer@mpimet.mpg.de)

**Abstract.**

Dynamic vegetation models simulate global vegetation in terms of fractional coverages of a few plant functional types
(PFTs). Although these models often share the same concept, they differ with respect to the number and kind of PFTs, complicating the comparability of simulated vegetation distributions. Pollen-based reconstructions are initially only available in form of time-series of individual taxa that are not distinguished in the models. Thus, to evaluate simulated vegetation distributions, the modelling results and pollen-based reconstructions have to be converted into a comparable format. The classical approach is the method of biomisation, but hitherto, PFT-based biomisation methods were only available for
individual models. We introduce and evaluate a simple, universally applicable technique to harmonize PFT-distributions by assigning them into nine mega-biomes that follow the definitions commonly used for vegetation reconstructions.

The method works well for all state-of the art dynamic vegetation models, independent of the spatial resolution or the complexity of the models. Large biome belts (such as tropical forest) are well represented, but regionally confined biomes (warm-mixed forest, Savanna) are only partly captured. Overall, the PFT-based biomisation is able to keep up with the
conventional biomisation approach of forcing biome models (here: BIOME1) with the background climate states. The new method has, however, the advantage that it allows a more direct comparison and evaluation of the vegetation distributions simulated by Earth System Models. Thereby, the new method provides a powerful tool for the evaluation of Earth System Models in general.

## 1 Introduction

Within dynamic global vegetation models (DGVM), the natural vegetation distribution is usually represented in the form of plant functional types (PFT), i.e. plants are grouped with regard to their physiology and physiognomy (Prentice et al. 2007). These PFTs differ with respect to phenology, albedo, morphological and photosynthetic parameters and are usually constraint by an individual bioclimatic range of tolerance defined by temperature thresholds. These thresholds mimic the cold resistance, chilling and heat requirements of the plants and determine the area where the PFTs can establish.

In most DGVMs, a 'mosaic' approach is used, i.e. each grid-box of the land surface is split into separate parts for a non-vegetated and a vegetated fraction that is further tiled in mosaics, taking sub-grid scale heterogeneity into account. Thus, several PFTs can cover the same grid-cell and compete for space via their net primary productivity (e.g. Sitch et al., 2003; Krinner et al., 2005; Reick et al., 2013). Non-vegetated area (seasonally bare soil or permanently bare ground) is produced where plant productivity is too low.

Although the main principles for the calculation of PFT-distributions are similar among most DGVMs, they vary regarding the number and kind of PFTs used to represent the global vegetation. Natural PFTs range from two in e.g. VECODE to ten in e.g. LPJ and ORCHIDEE (Tab.1). Even within the same model, PFT variety can differ between individual simulations. Some include land-use, some do not. In some models the natural PFTs can also deviate (e.g. LPJ). These differences among the simulations and models prohibits the inter-model comparability of simulated global vegetation distributions and the

comparability with pollen-based reconstructions. The latter originally display the vegetation in form of taxa compositions. Therefore, vegetation simulations and reconstructions need to be converted into a compatible format.

In the last two decades, taxa to PFT assignment- and biomisation-methods for pollen-based reconstructions have been developed (e.g. Prentice al. 1996, Ni et al., 2010, Harrison et al. 2010) and pollen-based biome syntheses have been provided (Prentice et al., 1998 and 2000, Bigelow et al., 2003; Ni et al. 2010, Harrison et al., 2017, Tian et al. 2017). These

reconstructed biome maps have been used extensively to evaluate simulated biome distributions obtained from diagnostic biome models (e.g. Prentice et al., 1992, Haxeltine and Prentice, 1996, Kaplan et al., 2003) that have been forced with palaeoclimate fields simulated by General Circulation Models (e.g. Jolly et.al, 1998; Harrison et al., 2003; Wohlfahrt et al., 2008; Harrison et al., 2016, Dallmeyer et al., 2017). Using this method, fundamental palaeo-vegetation analysis can be undertaken, but simulated vegetation distributions resulting from the dynamically coupled vegetation models as part of the

General Circulation models are disregarded.

Several model studies have taken up this problem by introducing methods for biomising simulated PFT-distributions. Schurgers et al. (2006) derive biomes maps for the Eemian and mid-Holocene from the relative fractional coverage of the individual PFTs and the soil temperature, both simulated by LPJ. With this method, reconstructed major biome shifts could be reproduced. Roche et al. (2008) used the dominant PFT and the bioclimate limits defined in the biome model BIOME1

(Prentice et al., 1992) to biomise PFT cover fractions for the Last Glacial Maximum (LGM) simulated by VECODE. As VECODE distinguish as main PFTs only trees and herbaceous plants, not all biome types defined in BIOME1 could be considered (e.g. no shrubs). The computed biome map shows reasonable agreement with LGM land cover reconstructions. A similar approach was chosen by Handiani et al. (2011 and 2012) for calculating biome distributions at Heinrich Event 1, based on PFT simulations of TRIFFID and the CLM-DGVM. As these models strongly deviate in their PFT classification,

they applied different methods for biomisation. For TRIFFID, they first calculated the dominant PFT in each grid cell following the method by Cruxifix et al. (2005) and afterwards used temperature limitation defined in BIOME4 (Kaplan et al., 2003) to assign the dominant PFTs to mega-biomes. For CLM-DGVM, potential dominant PFTs were estimated by adopting the scheme of Schurgers et al. (2006) and biomes were differentiated with the help of temperature limitations that



follow the environmental constraints defined in CCSM3 (i.e. the fully coupled model used in their study, including the land and vegetation model CLM-DGVM).

Recently, Prentice et al. (2011) introduced another approach of biomising plant functional type distributions simulated by dynamic vegetation models. In their method, simulated foliage projective cover (FPC) is used to distinguish between desert, grassland/dry shrubland and forest biomes, which are further divided into forest and Savanna like biomes through the vegetation height. The assignment to e.g. boreal, temperate or tropical forest/Savanna (or parkland) is controlled by the tree-PFT composition. Climate limits are only used to distinguish the tundra biome. This method has successfully been used in several palaeo-vegetation studies (Kageyama et al., 2013; Calvo and Prentice, 2015) using different versions of the LPJ and ORCHIDEE models.

All of these methods have in common that they have been designed for individual models and hence need specific output not necessarily provided by all models. Therefore these methods can not directly be adopted for all existing dynamic vegetation models.

To harmonize (palaeo)-vegetation distributions simulated by dynamic vegetation models and thereby facilitate the evaluation of Earth System models and the comparison of model results and reconstructions, we developed a biomisation technique that is based on few input variables and simple differentiation rules. These include bioclimatic constraints using near-surface air temperature and assumptions on maximum required PFT coverage. We test this method on pre-industrial, mid-Holocene and Last Glacial Maximum vegetation simulations performed in nearly all state-of-the-art dynamic vegetation models. The skill of this biomisation approach is quantified via (standard) metrics, by comparing the converted biome maps with estimates of modern potential biome distributions (Ramankutty and Foley, 1999) and pollen-based reconstructions (Biome6000 database, Harrison, 2017).

## 2 Methods

### 2.1 Biomisation

The plant functional type (PFT) cover fractions simulated by the individual dynamic vegetation models are converted into nine different mega-biomes (Fig.1), using few bioclimatic limits and assumptions on the maximum required coverage of certain PFTs. The aggregation into the mega-biomes is in line with the definitions of the BIOME6000 project (cf. Harrison, 2017) that are also commonly used for grouping pollen-based reconstructions. Bioclimatic limits and the differentiation rules basically follow the biome assignment of the BIOME4 model (Kaplan et al. 2003). As input data, only climatological mean growing degree days, monthly mean 2m-air temperature, and multi-year mean PFT cover fractions are required. The limitation to few climatic rules and few variables needed enables the application of the method to all state-of-the art dynamic vegetation models.

In detail, the PFTs calculated by the respective dynamic vegetation model are assigned into the PFT groups 'desert', 'forest', 'wood', 'grass' and 'total vegetation'. If the model include land-use types, the affected areas are redistributed to the other PFTs



by simply scaling up the other PFT fractions proportionally to their ratio of the total natural vegetation. Based on these PFT groups, regions dominated by forest or wood (with the additionally constraint of the total vegetation cover exceeding 50%) are identified, as this is a necessary condition for the assignment of the land cover to forest biomes. Afterwards, the forest PFTs are split into boreal, temperate and tropical forest via temperature limits (see. Tab.2). If any of theses forest PFTs are

simulated directly in the vegetation model (e.g. in LPJ or ORCHIDEE), the original distributions are taken and the forest type is assigned to the dominant tree-type. The macro-PFTs are the first consistent vegetation classification shared by all input simulations, so that model-to-model comparison is also possible on this PFT level.

For the biomisation, the forest PFTs are considered first, i.e. regions in which forests or woody PFTs (for temperate and boreal forests) are dominant are assigned to tropical, temperate and boreal forests according to the macro-PFTs. The

temperate forest is further divided into warm-mixed forest and temperate forest with the help of the growing degree days distribution (Tab.3). The remaining area is then tested for fulfilling the constraints for the non-forest biomes. First, the Savanna and dry woodland region is identified by bioclimatic limitations (GDD5 > 1200°C and Tmin > 17°C ) and a woody coverage of at least 25%. The remaining vegetated area is assigned to the biome 'grassland and dry shrublands', if GDD0 exceeds 800°C or to the biome 'tundra', if GDD0 is below 800°C. The non-vegetated area, i.e. regions in which the

total vegetation cover is less than 20%, is either assigned to warm or to cold desert, depending on whether the annual mean temperature is above or below 2°C. For the biome 'Tundra', only 10% vegetation cover is needed.

We are aware of the simplicity of this approach, calculating the tundra and the grassland and dry shrubland biomes as residuum of the non-forested area, not directly depending on the simulated grass PFT-fraction. We decided to attribute main priority on the forested biomes as this is also the strategy commonly used in DGVMs and Biome Models.

To assess the performance of the biomisation based on simulated PFTs, we additionally biomise the underlying climate which is the conventionally used procedure of biomising global climate model output. For this purpose, we use the biome model BIOME1 (Prentice et al., 1992) that calculates the biome distribution in equilibrium to the input climate. As forcing, BIOME1 needs the monthly mean climatological precipitation, near-surface temperature and cloudiness. The original biomes has been grouped into the same mega-biome classification that is used for the PFT-based approach.

**2.2 Simulations**

Simulations from nearly all state-of-the-art global dynamic vegetation models that are included in Earth System models have been selected for biomisation. Overall, eight simulations for the pre-industrial climate (PI) and vegetation, four for mid-Holocene (6k) conditions and five for Last Glacial Maximum (LGM) conditions have been used (Tab.4). Most of these simulations were performed within CMIP5/PMIP3 under strict simulation and output protocols enabling direct comparison

between the models (Braconnot 2011, Taylor et al., 2012). These include the models MPI-ESM-P, IPSL-CM5A-LR, MIROC-ESM and HadGem2-ESM.

The MPI-ESM-P (Giorgetta et al., 2013) simulations have been performed at the Max-Planck-Institute for Meteorology and include the land model JSBACH with dynamic vegetation module (c.f Reick et al. 2013). In the pre-industrial control





simulation, vegetation pattern and land use were prescribed. For the palaeo-simulations, we use the simulations with
interactive vegetation. The spatial resolution for the atmosphere and land is T63 (i.e. approx. 1.875° on a Gaussian grid).
These simulations are referred to as MPI-ESM-T63 in the following. In a similar model setup, additional PMIP3-like
experiments have been undertaken for PI and LGM by Klockmann et al. (2016) in a coarser spatial resolution (T31, i.e.
approx. 3.75° on a Gaussian grid, MPI-ESM-T31).

IPSL-CM5A-LR (Dufresne et al. 2013) is the low resolution CMIP5 model version of the Institute Pierre Simon Laplace and
contains the terrestrial biosphere model ORCHIDEE (Krinner et al. 2005) that is run offline, forced with climate input. In the
PMIP3-simulations, vegetation and land use were prescribed. For better comparison with the gridded reference dataset (see
next section), the climate and PFT-fields have been interpolated bilinearly to a Gaussian T31 grid. Using the simulated LGM
climate of the PMIP3 simulation, Zhu (2016) performed additional experiments for LGM with ORCHIDEE-MICT
(Guimberteau et al. 2018), a model version with improved vegetation dynamic in the high northern latitudes (Zhu et al.
2015). The corresponding piControl simulation has been forced by CRUNCEP v.5.3.2 data
(http://dods.extra.cea.fr/store/p529viov/cruncep/V4_1901_2012/readme.htm). These simulations have been interpolated to a
Gaussian T63 grid.

HadGEM2-ESM (Collins et al. 2011) is the Earth System Model of the Met Office Hadley Centre and includes the
vegetation model TRIFFID (Cox, 2001). In all simulations used here, the model ran with interactive vegetation. The
piControl simulations include land-use types. The simulations have been remapped to a Gaussian T63 grid.

The dynamic vegetation model LPJ (Lund-Potsdam-Jena model, Stich et al., 2003) is usually used for offline simulations,
forced by climate simulations or observations. The simulations used here has been conducted in a similar model-setup as
described in Kleinen et. al. (2010), but has been re-done on a new computer (T. Kleinen, personal communication). The pi-
Control simulation has been forced by observational datasets (CRU TS3.1, Harris et al., 2014), the 6k simulation by output
from the CLIMBER-2 model. Both simulations have been interpolated to a Gaussian T63 grid and are referred to as CLIM-
LPJ in the following..

MIROC-ESM (Watanabe et al., 2011) is the Earth System Model of the Japan Agency for Marine-Earth Science and
Technology, Atmosphere and Ocean Research Institute (The University of Tokyo), and National Institute for Environmental
Studies. It includes the dynamic vegetation model SEIB (Sato et al.,2007). SEIB is a gap model, not using the tiling
approach. The PFT distribution has been calculated in the post-processing for CMIP5 via the relative net primary
productivity of the vegetation categories. The piControl simulation includes land use. These simulations have been remapped
to a Gaussian T31 grid.

CLIMBER2 (Petoukhov et al., 2000) is an Earth System Model of intermediate complexity and contains the vegetation
module VECODE (Brovkin et al., 1997). The LGM and piControl simulations have been extra undertaken for this study (T.
Kleinen, personal communication) and are referred to as CLIMBER in the following. The CLIMBER output has not been
interpolated as the simulation ran with a too coarse resolution of 10° latitude x 51° longitude. To compare with the data and
the other models, the CLIMBER output was re-gridded to 10°x10° grid without interpolation.





The dynamic global vegetation model 'CLM-DGVM' as part of the Community Earth System model (Hurrell et al., 2013) is currently under re-development. No appropriate simulations could be provided.

For the pre-industrial time-slice, two of out of eight simulations (MPI-ESM-T63 and IPSL-ESM-T31) were performed with fixed vegetation distribution, but this has essentially no effect on the biomisation procedure. Therefore, we include these simulations in our analysis. Nevertheless, the PFT-based biome-distributions for these simulations are expected to fit better to the references than the other simulations.

We emphasize that this study is an introduction of a new biomisation method and not an evaluation of the different

vegetation models with respect to the skill of simulating biome distributions. For this purpose, the different vegetation models would have to be forced by the same climate state, which was not the case in this simulation ensemble.

### 2.3 Preparing the reference datasets

As reference, we use the estimated global potential natural vegetation map by Ramankutty and Foley (1999, referred to as RF99 in the following), which is a combination of modern satellite-based vegetation observations (i.e. the DISCover land

cover dataset) and the vegetation compilation prepared by Haxeltine and Prentice (1996) that has been taken for regions dominated by land use at present-day. The RF99 dataset is available at 5 minute resolution and distinguishes 15 different biome types that are similar to the mega-biome classification used here. Thus, most biomes could directly be assigned to the mega-biomes types (Tab.5). RF99 additionally includes the biome 'Evergreen/Deciduous Mixed Forest/Woodland' which is classified into temperate forest in warm regions and into boreal forest in colder regions. As temperature threshold we choose

the limit of growing degree days on a basis of 5°C being higher than 900°C derived from modern observations (University of East Anglia Climatic Research Unit Time Series 3.1, University of East Anglia, 2008, Harris et al., 2012). Likewise the Savanna biome had to be split up as RF99 includes temperate Savanna which is explicitly excluded in the definition of the Savanna mega-biome used in this study. The threshold for warm Savanna is a mean temperature of the coldest month exceeding 10°C (Limit for existence of C4 grass). To compare RF99 with the different model simulations, RF99 had to be

remapped to the model grids. We decided to use the spatial resolutions T31, T63 and also prepared a map for the downscaled CLIMBER output (10°x10°grid). Within each of this model grid cells, the dominant mega-biome type in the 5-minute-resolved RF99 data was taken for covering the RF99 grid-box in T31 or T63 or in the 10°grid. In more details, each grid-box on a T31 Gaussian grid contains 45*45 grid-cells of the 5-minute-resolved RF99 data. Within these 45*45 grid-boxes the fractional coverage of all mega-biomes is calculated and the biome with the highest fraction is chosen for covering the T31

grid-box. For T63 arithmetically 22.5*22.5 grid-cells form one T63 grid-cell. Here, we take 23*23 RF99 grid-boxes with one grid-box overlap to equally distribute the 5 minute grid-cells to the T63 grid. We are aware of the fact that the latitudes in the Gaussian grids are actually not equidistant, so that the remapped RF99 biome distributions are slightly stretched towards the poles, but this effect is marginal and is not expected to shift the main biome belts. To compare equal number of grid-cells, the reference data is cut by the land-sea masks used in the individual simulations to only include grid-cells being on land in both

data-sets.



As further reference data, we use pollen-based reconstructions that are available for the pre-industrial, mid-Holocene and Last Glacial Maximum time-slice within the Biome6000 database (Harrison, 2017). The biome reconstructions have been grouped into the mega-biomes according to the suggestions made by the Biome 6000 project.

As pre-industrial reference climate, climatological monthly mean data of the year 1901-1930 from the University of East

Anglia Climate Research Unit Time Series 4.00 (CRU TS4, University of East Anglia, 2017) has been taken. This is the earliest period available. The CRU TS4 reference climate has additionally been used as forcing for the BIOME1 model to provide a 'best guess' for the pre-industrial biome map. We assume that neither the biomisation of simulated climate states (i.e. the classical method) nor the biomisation of simulated PFTs can agree better with any reference than this biome distribution, derived with a highly tuned biome model and the best global climate observation available. Therefore, we use

the level of agreement between the CRU TS4 biome map and the RF99 or the Biome6000 reconstructions as target value for our new biomisation method. The reference biome distributions and the CRU TS4-based biome map are displayed in Fig.2.

### 2.4 Metrics

### 2.4.1 Kappa Statistic

The Kappa statistic (Cohan, 1960) is a widely used quantitative map-comparison technique that has often been applied for assessing the performance of vegetation simulations (e.g. Monserud and Leemans, 1992; Prentice et al., 1992; Diffenbaugh et al., 2003; Tang et al., 2009). The Kappa statistic not only includes the actual observed similarity ($p_0$) of two categorical maps, but also considers the expected agreement ($p_e$), i.e. the agreement by chance. For each pair of compared grid-cells (or a pair of grid-cell and site) taken from the reference and the simulated biome distribution, a confusion matrix is prepared

containing all combinations of referenced and simulated biomes. Based on this error matrix, the agreement for each

individual mega-biome is given by the following Eq. (1, taken from Tang et al., 2009):  $\kappa_i = \dfrac{p_{ii} - p_{i.r}\, p_{c.i}}{((p_{i.r} + p_{c.i})/2 - p_{i.r}\, p_{c.i})}$

$\qquad$ (1)

where $p_{ii}$ is the individual entry for biome i on the main diagonal of the confusion matrix and $p_{i.r}$ and $p_{c.i}$ are the row total and

the column total of each biome i, respectively. The overall agreement is derived by Eq. (2):  $\kappa = \dfrac{p_0 - p_e}{1 - p_e}$  (2)

with  $p_0 = \sum\limits_{i=1}^{n} p_{ii}$  and  $p_e = \sum\limits_{i=1}^{n} p_{i.r}\, p_{c.i}$ .  $\kappa$ ranges from 0 (not better than agreement by chance) to 1 (perfect agreement). We additionally use the thresholds suggested by Landis and Koch (1977), classifying a $\kappa$ below 0.4 into poor





agreement, values between 0.4 and 0.75 in fair to good agreement and values exceeding 0.75 into very good to excellent agreement.

### 2.4.2 Fractional Skill Score (FSS)

The standard Kappa statistic underestimates the similarity of maps sharing a similar biome distribution but being slightly offset from each other (Foody, 2002; Tang et al., 2009). This problem is usually overcome by using the Fuzzy Kappa statistic allowing for fuzziness in category and fuzziness in location (Hagen, 2003 and 2009), but the Fuzzy Kappa statistic is only applicable to assess the similarity of categorical maps and can not be used for single point to gridded-data comparison. Reconstructions only exist for single sites and usually indicate not only the local nor the regional vegetation, but may contain

a large extra-regional component, depending e.g. on the configuration (mainly the size) of the lake (Jacobsen and Bradshaw, 1981). A single grid-cell to point comparison is, thus, only partly meaningful, more advisable is the inclusion of the surrounding grid-cells of the sites. Therefore, we looked for a metric taking agreement in the neighbourhood into account (such as the Fuzzy Kappa statistic) that could easily be adapted to site to gridded-data comparison. We decided to use the fractional skill score (FSS, Roberts and Lean, 2008). While this method was initially developed and applied for expressing

the performance of precipitation forecasts (e.g., Gilleland et al., 2009; Mittermaier et al., 2013; Wolff et al., 2014), it has recently been successfully used for different hydrological patterns (Koch et al. 2017). We further adapted the FSS method to biome distributions. For each mega-biome type, the reference (ref) and simulation (sim) is truncated into a binary map, i.e. we construct 18 maps (9 for the reference, 9 for the simulation), in which the grid cell being covered by the respective mega-biomes are filled with the value '1' and all other grid-cells are assigned to the value '0'. Based on these maps, the mean

fractional coverage of the respective mega-biome within the neighbourhood $N_{ij}$ (3 grid-cells in each direction for T31, 6 for T63, 1 for 10°grid) of each cell is calculated for the reference and the simulation. Afterwards, the mean-square error (MSE) between the simulation and the reference fractions for each individual mega-biome is calculated and normalized by the MSE representing the worst-case agreement (MSEw), i.e. the MSE reflecting no similarity between the reference and the

simulation. The fractional skill score is then given by Eq.(3)    $FSS = 1 - \dfrac{MSE}{MSEw}$     (3)

with    $MSE = \dfrac{1}{N} \sum\limits_{i=1}^{N_i} \sum\limits_{j=1}^{N_j} \left[ ref_{ij} - sim_{ij} \right]$    and:    $MSEw = \dfrac{1}{N} \left[ \sum\limits_{i=1}^{N_i} \sum\limits_{j=1}^{N_j} ref_{ij}^2 + \sum\limits_{i=1}^{N_i} \sum\limits_{j=1}^{N_j} sim_{ij}^2 \right]$   ; N is the number of all neighbourhoods.

Following Robert and Lean (2008), we define the lowest skill by the $FSS_{ran}$ of a random biome distribution with the same fractional coverage as the observed one over the domain ($f_0$). Likewise, the target skill is given by the FSS that is reached for a uniform distribution of the observed biome fraction everywhere in the domain ($FSS_{uni} = 0.5 + f_0/2$). As $FSS_{ran}$ and $FSS_{uni}$

deviate between the individual biomes, we compare the relative FSS (rFSS) given by FSS -$FSS_{uni}$.



The total rFSS is calculated as mean of all individual mega-biome scores. The total and individual rFSS can ranges from ca. -0.5 (as good as a random distribution) to ca. 0.5 (perfect agreement), depending on the extent of the individual biomes. The skill to reach is zero for all biomes. For simplicity, we use as abbreviation for the relative FSS also just FSS.

### 2.4.3 Best Neighbour Score

Neither the FSS nor the Fuzzy Kappa statistic is in its original format applicable for the comparison of site-data vs. gridded data. For quantifying the similarity of simulated biome distributions and pollen-based reconstructions, we therefore implement a new metric following both methods called the best neighbour score (BNS), accounting for agreement in the neighbourhood of the record site and therewith being more tolerant for the position of the site. Within this metric, not only the grid-box locating the record sites are used for comparison with the records, but also the surrounding grid-boxes (3 grid-boxes in each direction for T31, 6 for T63, 1 for 10°grid). Similar to the Fuzzy Kappa statistic, the similarity in the neighbouring grid-cells is expressed by a distance decay function. We here choose a Gaussian function (Eq.(4)), giving grid-cells directly at the site proportional larger influence than grid-boxes far away.

$$w = e^{\frac{-1}{2}*\left(\frac{distance}{3}\right)^2} \quad \text{with} \quad distance = \sqrt{dlon^2 + dlat^2} \qquad (4)$$

The best neighbour is defined as the nearest grid-box within the neighbourhood agreeing with the reconstructed biome type. The agreement for each record is then given by the distance weight of the best neighbour in each neighbourhood. It is equal to 1 if the grid-box locating the site indicate the same biome as reconstructed and it is equal to 0 if all grid-cells in the neighbourhood disagree with the record. The BNS is the mean of all individual neighbourhood scores. In contrast to the Fuzzy Kappa statistic, the BNS neither takes agreement by chance into account nor considers potential spatio-temporal autocorrelation.

### 3 Results

#### 3.1 Comparison of the PFT-based and climate-based biome-distributions for the pre-industrial time-slice

For the pre-industrial time-slice, the PFT coverage of eight different Earth System Model simulations has been converted into mega-biome distributions (Fig.3). Additionally, the underlying pre-industrial climate states are used as forcing for the BIOME1 model (i.e. the classical way of biomisation) to calculate the mega-biome distributions in equilibrium with the simulated climate states (Fig. 4). Overall, the PFT-based biome maps look similar to the climate-based ones. All major biome belts can be reproduced using the new method, independent of the resolution or the complexity of the vegetation models. The biomisation based on the PFT coverage generally assigns more grid-cells to forest or woody biomes (e.g. Savanna instead of grassland or desert) than the classical method. This is most noticeable in South America, where the area covered by tropical forests is strongly increased in the PFT-based biome-distribution, being more in line with observations. Likewise, the



Savanna and/or forest biomes are more spread on the African continent for nearly all simulations with the exception of CLIMBER and IPSL-ESM-T63.

The Asian forest regions are slightly larger in most PFT-based biome-distributions compared to the climate-based ones. This impression is reinforced by the fact that for CLIM-LPJ and IPSL-ESM-T63 the PFT-method suggests a pronounced boreal
forest belt in Northern Asia, not only reducing the size of the grassland but also of the temperate forest area.

For North America, the PFT-based approach yields less forest for MPI-ESM-T63 and IPSL-ESM-T31 than shown by the climate-based biomisation. As a consequence, the North American prairie is better represented in the PFT-based biome distributions. In Alaska, parts of the tundra regions suggested by BIOME1 tend to be replaced by boreal forest when using the new approach, which is generally more consistent to the observed vegetation.
The differences between the PFT-based and climate-based biome distributions can be caused by deficiencies in the biomisation methods, biases related to the imperfect vegetation models, or biases in the simulated climate. While the effect of shortcomings in the vegetation models can not be disentangled, the caveats of the PFT-based method and the effect of climate biases on the PFT-to-biome conversion are further discussed in Section 4.

**3.2 Quantitative comparison of the PFT-based biome-distributions with reference biome maps**

To quantify the skill of the new method to represent the global biome distribution, we compare the resulting biome maps with the modern potential natural vegetation cover estimated by Ramankutty and Foley (1999, RF99 in the following) and the pre-industrial biome reconstructions provided by the Biome 6000 project (Harrison et al., 2017). As target for the skill, the level of agreement between the BIOME1 derived biome-distribution of the observed climate (CRU-TS4, years 1901-
1930, cf. Fig.2) and the references data-sets are taken, i.e. κ of 0.68 and FSS of 0.13 (Fig.5) with respect to RF99 and a κ of 0.46 and BNS of 0.73 with respect to the reconstructions.

The PFT-based biome distributions agree well with the references, independent of the model. The Kappa statistic shows an overall agreement to RF99 between 0.54 (MIROC-ESM) and 0.79 (MPI-ESM-T63) revealing a good- to very good agreement (Tab.6 and Fig.5). Likewise, all models reach in total the level of good skill in the FSS metric (0.01-0.27). This
agreement is in line with and, for the simulation performed with prescribed vegetation (MPI-ESM-T63 and IPSL-ESM-T31), even better than the match between RF99 and the CRU-TS4 map that is taken as target skill. In addition, the skill is in the range of the values reached for the climate-based biome-distributions (Fig.5), though the spread between the individual models is larger for the PFT-based method. For the MPI-ESM simulations, the IPSL-ESM-T31 and partly the CLIMBER simulations, the PFT-based biomisation agrees even better with RF99 than the climate-based ones.
As expected, the Kappa statistic indicates that the PFT-based biome maps compare worse with the reconstructions than with RF99, underestimating the similarity of the biome maps and the point reconstructions. κ ranges from 0.2 (poor) to 0.49 (fair), which is in line with the target skill and the metrics for the climate-based biome maps. The BNS additionally considering



accordance in the neighbouring grid-cells of the record sites reveals a good to very good agreement of the PFT-based biome distributions and the records (between 0.40 and 0.74), not much lower than the target skill and in accordance with the
climate-based biomisations. For the MPI-ESM, IPSL-ESM-T31, HadGEM2-ESM and CLIMBER simulations, the PFT-based method even produces biome distributions that fit better to the reconstructions than the climate-based biome maps.

Despite the overall agreement between the PFT-based biome-distributions and the references, a closer look at the representation of individual mega-biomes in the converted maps indicate large differences among the models as well as among the individual mega-biomes (Fig.6). While tropical forests and deserts compares best with RF99, the biome warm-
mixed forest is not reproduced, independent of the underlying model simulation. This is partly related to the relatively coarse resolution of the models in comparison with RF99 that had to be interpolated to the coarse grids. Warm-mixed forests are small and rather patchily distributed and are thus rarely dominant in the coarse grid-cells. The coarser the grid is the more warm-mixed forest regions get lost during the interpolation. Therefore, the warm-mixed-forest biome is generally better represented for models using a higher spatial resolution (i.e. MPI-ESM-T63, CLIM-LPJ and IPSL-ESM-T63). The skill for
simulating the other biomes is very different for the diverse models. κ spreads from poor for one model to very good for other models. Correcting the PFT distribution in land-use areas by redistributing the area fraction to the other tiles has no impact on the performance of the method. The biome maps based on simulations applying land-use do not compare worse with RF99 than the maps of other simulations. Likewise, the complexity of the vegetation model and the number of distinguished PFTs has no significant effect on the representation of the biome distribution, indicating that the climate limits
used in the biomisation procedure are appropriate for the assignment of the PFTs to the distinct macro-PFTs. The differentiation of the PFT types (e.g. the different forest types) in vegetation models is often based on similar climate limits, regardless of whether the model is a complex dynamic vegetation model, or a simple biome model. With the exception of the PFT-based biomisation for CLIMBER, in which the coarse grid is clearly disadvantageous for capturing the reconstructed desert belts and the rather regionally confined biomes (Savanna, warm-mixed forest), the spatial resolution of the models is
not the primary factor for the spread in the metrics. The PFT-based method performs equally well for simulations using T63 as for simulations using T31 (in total), only the regionally distributed warm-mixed forest is better represented in finer resolutions.

In general, the skill in representing the individual mega-biomes is similar for the PFT- and the climate-based method. Both approaches have the same strengths and weaknesses, but the spread between the models is larger for the PFT-based
biomisations. In comparison with the classical method, the tropical, the warm-mixed and the boreal forest biomes tend to be slightly better represented by the PFT-based method. In contrast, the temperate forest, Savanna and grassland distribution - averaged over all models – fit better to RF99 when using the climate-based approach, although for individual simulations, κ derived for the PFT-biomisation exceeds the climate-based one. The Savanna and grassland biomes are particularly misrepresented in the biome maps that are based on the PFT-distributions simulated by MIROC-ESM, CLIMBER, CLIM-
LPJ and HadGEM2-ESM. The temperate forest is poorly reproduced only in the PFT-based biomisation of CLIM-LPJ.



Overall, the metrics indicate that the PFT-based method works similarly well as the classical approach of biomising climate states.

**3.3 PFT-based biome-distributions for the mid-Holocene and Last Glacial Maximum time-slice**

The sensitivity of the PFT-based method to changes in the vegetation cover is assessed by evaluating palaeo-biome
distributions. For the mid-Holocene time-slice, four different simulations have been analysed. The main vegetation change described by reconstructions are the northern shift of the northern hemisphere forest-belts, in particular a northward displacement of the Taiga-Tundra boundary, and the decrease of the desert areas compared to pre-industrial (Fig.7). According to the records, grassy vegetation reached at least up to 26°N at 6k, far into the modern central Sahara. For none of the models, this biome shift is reproduced, neither in the PFT-based (Fig.7) nor in the climate-based biome distribution (not
shown). The mean Sahara desert border shifts northward by one to two grid-cells in the biomisations (i.e. ca. 1.875° to 3.75°, Tab.7). This shift collocates with substantial reductions in the desert fractions simulated by the individual ESMs (Fig.8). Only in MPI-ESM-T63, vegetation is increased in the entire western and central Sahara, but this increase is lower than 20%, not leading to a change in the biome assignment from desert to grassland. As the climate-based biomisations performed with BIOME1 reveal a reduction of the Sahara desert area in the same magnitude as the PFT-based ones, we conclude that the
new biomisation method shows a reasonable sensitivity to the simulated changes in the desert fractions.

For all models with the exception of MIROC-ESM, the PFT-based biomisation reproduces an increased forest biome fraction in Eurasia north of 60°N during the mid-Holocene compared to pre-industrial, in line with the reconstructions (Tab.8). Though the magnitude of the change differs between the models, ranging from 0% within MIROC-ESM to 12% within CLIM-LPJ. For nearly all models (except for MIROC-ESM), the expansion of the forested area in the high northern latitudes
seen in the PFT biomisation is of similar magnitude as in the climate-based biomisation, confirming that the method covers past vegetation changes with reasonable sensitivity.

Overall, the biome distributions for the mid-Holocene compare equally well to the reconstruction as for the pre-industrial time-slice (Fig.9). Although, κ is similarly low, ranging from 0.17 in CLIM-LPJ to 0.38 in MPI-ESM-T63 (poor agreement), the spread in the models and the differences in κ between the PFT-based biomisation and the climate-based biomisation are
nearly identical to the results for the pre-industrial biome distributions. In line with the results for PI, the BNS indicate a good to very good agreement to the reconstructions (ranging from 0.44 in MIROC-ESM to 0.72 in MPI-ESM-T63). The skill to capture the reconstructed individual mega-biomes strongly depends on the number of available pollen records, thus, temperate and boreal forests are represented best (Fig.7), while the simulated Savanna regions are not supported by the reconstructions.

For the Last Glacial Maximum time-slice, five different simulations have been analysed. The main reconstructed vegetation differences at LGM compared to PI area strong equator-ward retreat of the forest biomes and an expansion of tundra and steppe regions, so that e.g. Europe was mostly covered by grassy biomes (Fig.10). The northernmost record indicating boreal



forest during LGM is located at approx. 51°N in Asia. The PFT-biomisations mostly reproduce this reduction and the shift in Northern Hemisphere forest biomes (Fig.10), though the extent of the shift is underestimated. Forest reaches up to 50°N-

(CLIMBER) to 65°N (MPI-ESM-T63). The boreal forest position for MIROC-ESM is not much changed compared to PI, but boreal forest nearly replace the temperate forest biome.

The overall agreement of the PFT-based biome distributions with the reconstructions is rather fair, but in line with the results for the climate-based distributions. κ ranges from 0.07 in MPI-ESM-T31 to 0.23 in CLIMBER, only indicating a poor similarity of the biome maps and records (Fig.11). The BNS ranges from 0.24 (MIROC-ESM) to 0.57 (MPI-ESM-T63)

revealing a fair to good agreement. The values for both metrics are in the same magnitude as for the climate-based biomisation. Similar as for the PI time-slice, neither the complexity nor the spatial resolution are the main reason for the differences between the simulations. The spread in the skill of representing the individual biomes is large, and no systematic bias for one model can be found. With the exception of the biomisation for CLIMBER, the Savanna biome is misrepresented for all simulations, independent of whether the PFT-based  or the climate-based method was used. Within the model

ensemble, tropical and temperate forest can be reproduced best.

## 4. Discussion

### 4.1 Caveats in the method

Even if the biomisation is restricted to mega-biome level, no clear definitions exist to distinguish biomes in terms of plant functional type compositions. While the bioclimatic limits used in the biome models are based on empirical analysis, no

equivalent classification regulates the biomisation of PFTs. We particularly face this problem in finding a meaningful threshold of maximum tree cover needed for defining forests. When is an accumulation of trees identified as forest? As models tend to underestimate the forest coverage and forest extent in the high northern latitudes (cf. Loranty et al., 2013)  we choose the assumption of tree cover being just dominant in forested grid-cells, although this limit is very low. We test other limits (e.g. absolute dominance, i.e. fractional coverage exceeding 50%), but these works worse, for most simulations used in

this study as well as other simulations.

The biome 'warm-mixed forest' (subtropical forest) is only vaguely defined as it shares most tree species with tropical evergreen broadleaf forests (cf. Ni  et al. 2010). These biomes tend to overlap so that their differentiation in reconstructions is very difficult (Chen et al. 2010). Therefore, warm-mixed forests might be misrepresented in the reference data leading to ambiguous evaluation of the biomisation method. This is further hampered by the rather regional distribution of the warm-

mixed forest covering only few grid-cells. Thus, the correct reproduction of the warm-mixed forest distributions is very challenging. As biome models such as BIOME1 generally manage to simulate warm-mixed forest at the correct locations, we adopt the bioclimatic limits from the biome models for defining warm-mixed forest. Though, in biome models and also in the dynamic vegetation models used in this study, warm-mixed forest is based on temperate broadleaved trees, inconsistent

with the definition used for reconstructions. This is related to the limited number of PFTs in the models grouping a large

number of species into few PFTs.

Furthermore, not all biomes can be differentiated by the structural composition or climatic tolerance. The biome 'Savanna' is the second-largest ecosystem in the tropics, covering approx. one fifth of the global land surface (Scholes and Hall, 1996). It occurs in climatic zones that are also suitable for forest and grasslands (Lehmann et al., 2011) and is, thus, very variable regarding the plant composition. Tree fraction can vary from very dense (open forest Savanna) to nearly zero (Torello-

Raventos et al., 2013). While Savannas require the coexistence of trees and C4-grass, they can only be distinguished from forests by their unique functional ecology, fire tolerance and shade intolerance (Ratman et al., 2011). These features make Savannas unstable and vulnerable to changes in e.g. grazing, fire regime and climate, transforming Savannas into forest or grasslands (Franco et al., 2014). The functional diversity of Savannas is not adequately included in DGVMs nor considered in the biomisation method presented here. As even C4-grass is not simulated in all models, we had to define the Savanna

biome very rudimentary by a mixture of wood and grass and by bioclimatic limits, i.e. a mean temperature of the coldest month exceeding 10° which is taken as limit for C4-grass in dynamic vegetation models (e.g. JSBACH; c.f. Reick et al., 2013). The Savanna biome might therefore not be represented well. At least in the palaeo-simulations, most biomisations do not capture the reconstructed Savanna area, but this may also be related partly to the fact that only few records exist indicating Savanna during LGM and 6k.

Similar to dynamic vegetation models, the priority in the biomisation procedure is given to forest biomes. It is first tested, whether forest biomes are suitable for covering the grid-cell, before Savanna is distributed. Grasslands and Tundra are assigned to the residual grid-cells, independent of the real grassy PFT cover fractions. The only restriction is a total vegetation coverage exceeding 10% for Tundra or 20% for grassland to be distinguishable from deserts. This method has the large disadvantage that biases in the forest distribution propagate throughout the assignment of all biomes with the exception

of deserts. The forest biome distribution calculated for the different models is further tested in Section 4.3 for the pre-industrial time-slice.

Another problem is the inclusion of anthropogenic plant functional types in some simulations, making the biome distribution less comparable to the reference data. Although land-use is often prescribed in the models, this process cannot be reversed in the final output data. The area chosen for land-use is historically determined and is based on human decisions and not

primarily on climate conditions. These human pathways cannot be reproduced in simple biomisation methods nor in the current dynamic vegetation models. We artificially rescale the natural vegetation in human-affected regions by redistributing the fraction of anthropogenic PFT coverage proportionally to the natural PFTs. This is a very simple approach and only partly in line with the implementation of land-use in the dynamic vegetation models. For instance, within JSBACH pasture is preferentially assigned to natural grasslands, forests are only affected if prescribed pasture fraction exceeds the natural

grassland area (cf. Reick et al., 2013). This rule is plausible, but not reversible and therefore not appropriate for the biomisation method presented here. The results show that biome maps based on models including land-use do not agree





worse with the references than the other simulations, underlining that the redistribution method used here provides a good approximation of the natural vegetation cover.

The method of PFT-biomisation basically uses bioclimatic limits that have been inferred for the modern vegetation-climate

relationships. These limits may not be valid for all time periods. During LGM, the atmospheric $CO_2$-concentration was substantially lower than today, which may also affect the response of the plants to the background climate. Likewise, the procedure of reconstructing palaeo-vegetation often include modern analogue techniques. This may lead to biases in the modelling results and the reconstructions taken as reference. The κ values for the LGM time-slice were quite low, indicating disagreement between the simulated and reconstructed biome distributions.

A rather technical problem is the  interpolation of the PFT-distributions to the T31 or T63 grid that partly leads to a decrease in the global area to be compared with the reference datasets due to a mismatch of the land-sea-masks. In regions with a strong change of the PFT fractional composition (e.g. desert border, coastal region), the interpolation may produce blurry transitions in the PFT distributions resulting in an erroneous conversion into the mega-biomes.

### 4.2 Biases in the pre-industrial biome distributions and the influence of the climate background

The classical method of biomising climate states and the new PFT-based method result in similar biome-distributions for most models and all time-slices. Generally, the PFT-based method produces more forest in comparison with the classical approach. This is mainly related to the rather low limit of forest fraction needed in the assignment of forest in the PFT-biomisation procedure. In forest regions, where Earth System Models tend to produce large biases in the climate state, the PFT-based approach may be more suitable for the biomisation. Therefore, the tropical, warm-mixed and boreal forest are

probably better represented by the PFT-method. However, the biomisation of the PFT-distributions itself strongly depends on the underlying climate, affecting both the differentiation into the biomes as well as the simulation of the PFT coverage in the different dynamic vegetation models. To accurately compare the performance and the skill of the different vegetation models to represent biome distributions, the models should be forced by the same climate state, but only few models can be run offline. Therefore, this study is not thought as model evaluation, but as introduction to the biomisation method and as test

whether the procedure works for models of different complexity and simulations for different time-slices.

To assess the contribution of the effect of biases in the underlying climate to the differences in the PFT-based biome distributions among the models and between the models and the references, we compare the pre-industrial climate-based biomisations with the CRU-TS4 dataset. A sensitivity study is performed following Dallmeyer et al. (2017) to relate differences in the biome distributions to precipitation or temperature deviations in the background climate (Fig.12).

Generally, PI biome disagreement in the high northern latitudes is associated with biases in temperature while disagreement in low latitudes co-occurs with precipitation biases.

The similarity of the converted PI biome distributions and RF99 is lowest for CLIMBER, MIROC-ESM and CLIM-LPJ. While for CLIMBER the coarse resolution (i.e. the very different land-sea masks) may be the main responsible factor for

disagreement, total κ is reduced by an underestimation of grasslands and Savanna and an overestimation of the forests for
MIROC-ESM when using the PFT-method. This is exactly opposite to the biases occurring in the climate-based biomisation,
indicating that the climate is not the primary origin of the differences. For this specific model, the PFT-based biomisation
strongly differs from the climate-based one. This may at least partly be related to the handling of vegetation in the model.
The vegetation model 'SEIB' included in MIROC-ESM is a gap model, not using the tiling approach (Sato, 2007). PFT
fractions have only been estimated during the CMIP5 post-processing, based on the NPP ratios of the different vegetation
categories. This approach might lead to an overestimation of forest. On the other hand, according to BIOME1 the
underestimation of the tropical forest domain in the climate-based biomisation for MIROC-ESM is caused by the way to dry
climate in South America.

For the PFT-biomisation of CLIM-LPJ, the low score is basically caused by an overestimation of boreal forest at the expense
of temperate forest and an underestimation of the Savanna regions. Both errors are mainly not climate driven. Within
dynamic vegetation models explicitly calculating boreal and temperate forest, these forest types can coexist. To give a clear
assignment, we decided to differentiate both forest types by the dominant tree PFT, i.e. if the boreal tree fraction exceeds
temperate tree fraction, forest fraction is assigned to boreal forest and vice versa. This partly disagrees with the handling in
biome models, as e.g. in cool mixed forest, boreal trees could be the dominant PFT and temperate trees only the sub-
dominant PFT (c.f. Kaplan et al., 2003), but this biome would be assigned to the mega-biome 'temperate forest'. We assume
that due to a slight overestimation of boreal forest coverage in Europe and at the modern boreal to temperate forest transition
zone within CLIM-LPJ, the vegetation in these regions is grouped into the mega-biome 'boreal forest'. In South America,
tropical forest fraction is overestimated by CLIM-LPJ with values exceeding 80% in most regions of Brazil, precluding the
Savanna biome. Within North Africa, CLIM-LPJ simulates hardly any regions with coexisting substantial forest and grass
fractions. Either tropical trees are clearly the dominant PFT (assigned to tropical forest) or forest fraction is too low (below
10%) to be assigned to Savanna. The defined limits for Savanna are only fulfilled for very few grid-cells.

For MPI-ESM-T31, the boreal forest biome is strongly underestimated in the PFT-based biomisation. The BIOME1 results
clearly relate this bias to a too cold climate (GDD5 limit is not reached in BIOME1) which also affects the simulation of
trees in JSBACH sharing the same bioclimate limit. Therefore, forest fraction in MPI-ESM-T31 is underestimated for the
northern latitudes. IPSL-ESM-T31 shows a dry bias in South America resulting in a too low tropical forest biome cover, in
both, the climate-based and the PFT-based biomisation. BIOME1 reveals another systematic bias for the MIROC-ESM
simulation indicating too much temperate forest in North America at the expense of grassland and partly of boreal forests.
This overestimation of temperate forest is induced by a too wet climate favouring growing of trees and a rather too warm
climate in the high northern latitudes.

### 4.3 Evaluating the distribution of forest biomes

Due to the forest priority rule in the biomisation method, the skill of the models to represent the non-forested biomes
depends on how well the forest distribution can be reproduced. To further assess the performance of the method with respect



to the forest biomisation, we analyse the pre-industrial zonal mean forest fraction in form of the zonal sum of forested area per latitude to be independent of the different grid sizes used for the individual simulations (Fig.13). For nearly all models, the zonal forest fraction is underestimated in the high northern latitudes and the zonal maximum is shifted southward, although the defined limit of minimum required tree fraction is already quite low. This bias is most obvious for MPI-ESM-T31 and IPSL-ESM-T63. While for MPI-ESM-T31, the coexistence of the bias in both, the climate-based and the PFT-based biomisation, underlines the effect of the too cold climate on the forest distribution, the strongly shifted high latitude forest maximum in IPSL-ESM-T63 is probably not climate driven.

In the tropical regions, the models tend to underestimate the forest fraction when using the climate-based method whereas forest fraction is often too high in the PFT-based biomisations, probably related with the low tree fraction limit needed for forest assignment. The tropical forest fraction based on the CLIMBER climate and PFT-distribution is strongly overestimated. This is at least partly caused by the coarse grid and specific land-sea-mask used in the model.

To further quantify the biases in forest fraction, we compare the centred root mean square error (cRMSE), the Pearson correlation coefficient (r) and the zonal variability between the simulations and the RF99 reference, combined in a Taylor-diagram (Fig.14). Overall, the simulated zonal forest fraction agrees well with the reference. All biomisations show a good to nearly perfect pattern correlation with values exceeding 0.77, independent of the chosen method. For most models, the Pearson correlation coefficient even exceeds 0.9. ThePFT-based biomisation is worst for CLIMBER and MIROC-ESM, revealing a too large standard deviation and a cRSME of 0.83 and 0.73, respectively. For MPI-ESM-T31, spatial variability is slightly too low and the cRSME is 0.61 using the PFT-based method and 0.63 in the climate-based biomisation, reflecting the common underestimation of the boreal forest. As expected, best performance can be observed for the simulations with prescribed PFT coverage undertaken within MPI-ESM-T63 and IPSL-ESM-T31, sharing a similar standard deviation with RF99, a pattern correlation coefficient of 0.98 and a cRMSE of 0.21, which is even better as the biomisation of the CRU TS4 data (cRMSE of 0.28).

## 4.4 Comparison of the biomisation method with the approach of Prentice et al.

Prentice et al. (2011) introduced a biomisation method (further referred to as FPC method) that is fundamentally different to the method presented in this study. The assignment to the different biomes is controlled by the foliage projective coverage, the vegetation height and the PFT composition. Climatic limitations in form of growing degree days are only used to distinguish the tundra biomes.

Unfortunately, the foliage projective cover and the vegetation height are not included in the standard output of the vegetation models and are therefore not available for the simulations used here.

To compare both methods, we therefore only use the simulations performed within IPSL-ESM-T63, which has been biomised by Zhu (2016) following the approach of Prentice et. al. (2011). The biome output has been grouped into mega-biomes and remapped to a T63grid in the same way as the RF99 reference were prepared.



Beside the Savanna regions, the derived biome map resembles the map resulting from the PFT-based method (Fig.15).
Prentice et al. distinguish temperate parkland, sclerophyll woodland and boreal parkland that all have been assigned in the mega-biome Savanna, but Savanna is only defined as tropical Savanna in our method. This complicates the comparison of the biome maps and leads to strong differences in the Savanna distribution between both methods and between the biomisation using the FPC method and RF99, although we leave the temperate Savanna in RF99 for better comparison.

Since boreal parkland is not included in RF99, the PFT-method introduced here yield better results for boreal forest and
Savanna than the FPC method. Additionally, the warm forest is more appropriately reproduced. In contrast, temp. forest and grassland is better represented using the FPC method. All other biomes are equally well simulated for the PI time-slice.

The FSS metric (Tab.9) indicate that the PFT-based method (0.13) agrees in total slightly better with the reference than the other approach (0.10). Overall, the biomisation using the FPC method reaches a $\kappa$ of 0.59 (vs. 0.63 for the PFT-based method) compared to RF99 and 0.19 (vs. 0.24) compared to the Biome6000 pollen data. BNS is 0.53 for the FPC- method
and 0.55 for the PFT-method.

The LGM biome distribution can be captured slightly better using the FPC method (k=0.17 vs 0.13, and BNS=0.54 vs. 0.41). Particularly the tropical forest, desert and Savanna biome agree better with the reconstructions than for the PFT-based method, but at least for the latter biomes, the record density is very low which may distort the results.

**5 Summary and Conclusion**

Dynamic global vegetation models use different kinds and numbers of plant functional types to represent the global vegetation. These PFT distributions can neither be directly compared between different models nor between models and reconstructions, which were hitherto mostly provided in form of biomes. We therefore have developed a method for biomising simulated PFT distributions and have tested this method for nearly all state-of-the art dynamic global vegetation models based on simulations for the pre-industrial, mid-Holocene and Last Glacial Maximum time-slices.

Overall, the method works well for all models and can keep up with other biomisation techniques. The comparison with different references datasets (i.e. pollen-based reconstructions and estimates of the potential natural vegetation) reveals a similar agreement with the PFT-based biomisation than with biome distributions inferred from the biome model BIOME1 (Prentice, 1992) that has been forced with the background climates. The comparable skill to the BIOME1 model, which is tuned to represent the global vegetation as well as possible, is partly achieved by the use of bioclimatic limits that are in line
with the definitions in biome models.

The skill of capturing the global biome distributions is independent to the spatial resolution and the complexity of the vegetation model or the integration of land use. For models just using two different PFTs (CLIMBER) the method performs equally well as for models using ten different PFTs (e.g. IPSL-ESM). Only the very coarse resolution in the CLIMBER model hampers the comparability with the single point reconstructions, in particular for biomes with a very limited number



of available records. In addition, the quantitative comparison of the biomised vegetation distributions among each other and with the gridded reference data is complicated by the very different model resolutions.

In general, large biome belts (such as tropical forest) can be simulated best, while rather regionally confined biomes such as Savanna and warm-mixed forest are not as well represented. This may at least partly be related to the fact that these biomes can not be defined clearly via PFT cover fractions. Savannas are characterized by a distinct functional ecology and can not

be differentiated from other tropical biomes via plant composition or climatic tolerance. The warm-mixed forest biome shares many tree species with tropical forest leading to an overlap of theses biomes and uncertainties in the reconstructions. For the palaeo-simulations, the agreement between the individual mega-biome distributions derived by the PFT-method and the reconstructions strongly depends on the number of available records. The main vegetation differences between the pre-industrial and mid-Holocene or Last Glacial Maximum time-slices are captured by most models and are also reflected in the

PFT- and climate-based biomisation, indicating a reasonable sensitivity of the conversion method. In total, the Kappa statistic reveals only poor agreement between the simulated biomisations and the reconstructions for LGM, which might be related to the use of bioclimatic limits inferred from modern observations that may not be valid for climate states being totally different from present day's.

We have provided a simple but powerful method for the biomisation of simulated plant functional type distributions that requires only few input variables and can hence be applied to all kind of dynamic global vegetation models. The method can keep up with the classical biomisation approach of forcing biome models with climate states. As the biomisation of the simulated PFT-fractions indirectly accounts for all processes included in the dynamic vegetation models (e.g. ecophysiological response of the plants to changes in the environment such as atmospheric $CO_2$-level), the biomisation of

PFTs directly represents the output of the vegetation modules of an Earth System Model. The biomisation of the simulated vegetation, thus, facilitates the direct comparison between different Earth System Models, and between models and reconstructions. It is therefore a powerful method for the evaluation of Earth System Models, particularly suitable for the assessment of recent palaeo-vegetation changes.

**6 Code and data availability**

The PMIP3 simulations of MPI-ESM-T63, ISPL-ESM-T31, MIROC-ESM, and HadGEM2-ESM can be downloaded from the Earth System Grid Federation. Simulation IDs are listed in Table 4.

The tool for the biomisation of PFT-distribution, input data, other scripts used in the analysis and supplementary information that may be useful in reproducing the authors' work will be archived by the Max Planck Institute for

Meteorology and can be obtained by contacting publications@mpimet.mpg.de





**Acknowledgements**

This work contributes to the project PalMod, funded by the German Federal Ministry of Education and Research (BMBF), Research for Sustainability initiative (FONA, www.fona.de). AD was financed by PalMod. We thank J. Koch (GEUS) for fruitful discussion on metrics and T.Kleinen (MPI-M) for performing the CLIMBER and CLIM-LPJ simulations and his

helpful comments on an earlier version of this manuscript. We furthermore thank Dan Zhu (IPSL) for providing the IPSL-ESM-T31-simulations and the biome data derived by the FPC-biomisation method.

We acknowledge the World Climate Research Programme's Working Group on Coupled Modelling, which is responsible for CMIP, and we thank the climate modeling groups (listed in Section 2.2 of this paper) for producing and making available their model output. For CMIP the U.S. Department of Energy's Program for Climate Model Diagnosis and Intercomparison

provides coordinating support and led development of software infrastructure in partnership with the Global Organization for Earth System Science Portals.

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

855   .





| | JSBACH | LPJ | ORCHIDEE | DGVM/CLM | TRIFFID | VECODE |
|---|---|---|---|---|---|---|
| Tropical trees | Tropical broadl. evergreen | Tropical broadl. evergreen | Tropical broadl. evergreen | Tropical broadl. evergreen | | |
| | Tropical broadl. raingreen | Tropical broadl. raingreen | Tropical broadl. raingreen | Tropical broadl. raingreen | | |
| Extra-tropical trees | Extra-tropical evergreen | Temperate needlel. evergreen temperate broadl. evergreen | Temperate needlel. evergreen | Temperate needlel. evergreen | Broad-leaved trees | trees |
| | | | Temperate broadl. evergreen | Temperate broadl. evergreen | & | |
| | | Boreal needlel. evergreen | Boreal needlel. evergreen | Boreal needlel. evergreen | Needle-leaved trees | |
| | Extra-tropical deciduous | Temperate broadl. deciduous | Temperate broadl. deciduous | Temperate broadl. deciduous | | |
| | | Boreal needlel. deciduous | Boreal needlel. deciduous | Boreal deciduous | | |
| | | Boreal broadl deciduous | Boreal broadl. deciduous | | | |
| shrubs | Raingreen shrubs Cold deciduous shrubs | | | | shrubs | |
| Grass | C3 grass | C3 grass | C3 grass | Artic grass C3 grass | C3 grass | herbaceous |
| | C4 grass | C4 grass | C4 grass | C4 grass | C4 grass | |

**Tab.1: The plant functional types (PFT) used in the different state-of-the-art dynamic global vegetation models**




| PFT | Bioclimatic limitation |
|-----|------------------------|
| Boreal forest | GDD5 ≤ 900°C |
| Temperate forest | 900°C < GDD5 ≤ 3000°C |
| Tropical forest | Tcmin > 15.5°C |

**Tab.2 Bioclimatic limits used for splitting the forest PFTs simulated in the dynamic global vegetation models into the macro-PFT groups boreal, temperate and tropical forests. Taken are only temperature related variables, i.e. the growing degree days on a basis of 5°C (GDD5) and the monthly mean temperature of the coldest month (Tcmin). If the dynamic global vegetation model simulates any of these forest types directly, the simulated distribution is kept.**






| Mega-biome | minimum coverage needed | Bioclimatic limit |
|---|---|---|
| Tropical forest | Tropical forest dominant | Tcmin > 15.5°C |
| Warm-mixed forest | Temperate trees dominant | GDD5 > 3000°C |
| Temperate forest | temperate wood dominant | GDD5 ≤ 3000°C |
| Boreal forest | Boreal wood dominant | GDD5 ≤ 900°C |
| (warm) Savanna and dry woodland | Woody coverage > 0.25 | GDD5 > 1200°C, Tcmin > 10°C |
| Grassland and dry shrubland | Total vegetation cover > 0.2 | GDD0 ≤ 800°C |
| Tundra | Total vegetation cover > 0.1 | GDD0 < 800°C |
| Warm desert | Total Vegetation cover < 0.2 | Tann > 2°C |
| Cold desert / ice | Total Vegetation cover < 0.2 | Tann < 2°C |

**Tab.3 Bioclimatic limits and assumptions on minimum PFT coverage needed for the assignment of the macro-PFT-groups into the 9 Mega-Biomes. Similar as for the assignment into the macro-PFTs, only temperature-based limitations are used, i.e. the growing degree days on a basis of 5°C (GDD5) or on a basis of 0°C (GDD0), the monthly mean temperature of the coldest month (Tcmin), and the annual mean temperature (Tann).**





| Model acronym | Model /DGVM | Simulations | resolution | PFT | Reference |
|---|---|---|---|---|---|
| MPI-ESM-T63 | MPI-ESM-P JSBACH | PiControl* | T63 | 8+4 | cmip5.output1.MPI-M.MPI-ESM-P. piControl.mon.land.Lmon.r1i1p1.v20120315 |
| | | 6k | | | cmip5.output1.MPI-M.MPI-ESM-P. midHolocene.mon.land.Lmon.r1i1p2.v20120713 |
| | | LGM | | | cmip5.output1.MPI-M.MPI-ESM-P. lgm.mon.land.Lmon.r1i1p2.v20120713 |
| MPI-ESM-T31 | MPI-ESM-P JSBACH | piCTL, LGMref | T31 | 8 | Klockmann et al., 2016 |
| IPSL-ESM-T31 | IPSL-CM5A-LR ORCHIDEE CRUNCEP or | PiControl* | 1.875° x 3.75° | 10+2 | pmip3.output.IPSL.IPSL-CM5ALR.piControl. monClim.land.Lclim.r1i1p1.v20140428\| |
| IPSL-ESM-T63 | IPSL-CM5A-LR ORCHIDEE-MICT | | 2°x2° | 10 | Zhu (2016) |
| HadGEM2-ESM | HadGEM2-ES TRIFFID | PiControl* | 1.875° x 1.25° | 6+2 | cmip5.output1.MOHC.HadGEM2-ES. piControl.mon.land.Lmon.r1i1p1.v20111007 |
| | | 6k | | | mip5.output1.MOHC.HadGEM2-ES.midHolo cene.mon.land.Lmon.r1i1p1.v20120222 |
| CLIM-LPJ | CRU/Climber2 LPJ | | 0.5°x0.5 | 9 | similar as in Kleinen et al. 2010 |
| MIROC-ESM | MIROC-ESM SEIB | PiControl* | T42 | 8+2 | cmip5.output1.MIROC.MIROC-ESM.piControl. mon.land.Lmon.r1i1p1.v20120710 Watanabe et al., 2011 |
| | | 6k | | | cmip5.output1.MIROC.MIROC-ESM.midHolo cene.mon.land.Lmon.r1i1p1.v20120710 |
| | | LGM | | | mip5.output1.MIROC.MIROC-ESM.lgm. mon.land.Lmon.r1i1p1.v20120710 |
| CLIMBER | Climber2 VECODE | PI, LGM | 10°x10° | 2 | Kleinen (personal communication) |


**Tab.4: Overview of the simulations used for testing the biomisation method. Listed are the model acronym, the model name, the name of the included dynamic global vegetation model (DGVM), the simulations used in this study, the spatial resolution used in the simulations, the number PFTs (natural + anthropogenic) and the simulation reference. Simulations marked with (*) include land-use.**






| Biomes in Ramankutty and Foley (1999) | Mega-biomes |
|---|---|
| 1 Tropical Evergreen Forest/Woodland | Tropical forest |
| 2 Tropical Deciduous Forest/Woodland | |
| 3 Temperate Broadleaf Evergreen Forest/Woodland | Warm-mixed forest |
| 4 Temperate Needleleaf Evergreen Forest/Woodland | Temperate forest |
| 5 Temperate Deciduous Forest/Woodland | |
| 6 Boreal Evergreen Forest/Woodland | Boreal forest |
| 7 Boreal Deciduous Forest/Woodland | |
| 8 Evergreen/Deciduous Mixed Forest/Woodland | Temperate (GDD5 < 900) or boreal forest |
| 9 Savanna | Savanna and dry woodland, partly temperate forest |
| 10 Grassland/Steppe | Grassland and dry shrubland |
| 11 Dense Shrubland | Savanna and dry woodland |
| 12 Open Shrubland | Grassland and dry shrubland |
| 13 Tundra | Tundra |
| 14 Desert | (Warm) Desert |
| 15 Polar Desert/Rock/Ice | Polar desert / ice |

**Tab.5 Biome assignment of biome classes used in Ramankutty and Foley (1999) to the mega-biomes used in this study.**




| | κ vs. RF99 | κ vs. Records | FSS vs. RF99 | BNS vs. Records | pure fit | Fits in N (of 9117) |
|---|---|---|---|---|---|---|
| MPI-ESM-T63 | 0.65 | 0.42 | 0.15 | 0.67 | 43% | 7933 |
| MPI-ESM-T31 | 0.56 | 0.29 | 0.03 | 0.42 | 28% | 7254 |
| IPSL-ESM-T31 | 0.61 | 0.36 | 0.05 | 0.47 | 33% | 7373 |
| IPSL-ESM-T63 | 0.7 | 0.47 | 0.15 | 0.65 | 39% | 7865 |
| HadGEM2-ESM | 0.66 | 0.38 | 0.15 | 0.64 | 39% | 7755 |
| CLIM-LPJ | 0.7 | 0.40 | 0.21 | 0.66 | 39% | 8054 |
| MIROC-ESM | 0.66 | 0.35 | 0.11 | 0.41 | 27% | 6814 |
| CLIMBER | 0.62 | 0.23 | 0.00 | 0.26 | 26% | 3955 |
| CRU TS4_T63 | 0.68 | 0.46 | 0.13 | 0.73 | 50% | 8537 |
| CRU TS4 | - | 0.47 | - | 0.83 | 55% | 8686 |

**Tab.6: Metrics for the climate-based (BIOME1) biomisations (PI), i.e. κ for the comparison with the RF99 reference, κ for the comparison with the Biome6000 PI reconstructions, the relative fractional skill score (FSS), the best neighbour score (BNS), percentage of sites showing the same biome as the climate-based biomisation, and number of sites for which a grid-cell in the**
**neighbourhood (N) could be found showing the same biome as the site. The total number of records is 9117.**





| | PFT-based | | | Climate-based | | |
|---|---|---|---|---|---|---|
| | **PI** | **6k** | **6k-PI** | **PI** | **6k** | **6k-PI** |
| MPI-ESM-T63 | 17.72 | 21.45 | 3.73 | 15.85 | 19.59 | 3.74 |
| CLIM-LPJ | 15.85 | 17.72 | 1.87 | 17.72 | 21.45 | 3.73 |
| HadGEM2-ESM | 13.99 | 15.85 | 1.86 | 13.99 | 15.85 | 1.86 |
| MIROC-ESM | 16.7 | 20.41 | 3.71 | 16.7 | 20.41 | 3.71 |


**Tab.7 Position of the desert margin [° latitude] in North Africa at PI and 6k and the differences in position of the desert margin between 6k and PI [° latitude], for the PFT-based biomisations and the climate-based biomisations. The desert margin is here defined as latitude at which the zonal mean desert biome fraction averaged over the region 15°W to 30°E exceeds 50%.**






| | PFT-based | | | Climate-based | | |
|---|---|---|---|---|---|---|
| | PI | 6k | 6k-PI | PI | 6k | 6k-PI |
| MPI-ESM-T63 | 0.73 | 0.75 | 0.02 | 0.73 | 0.76 | 0.03 |
| CLIM-LPJ | 0.64 | 0.76 | 0.12 | 0.67 | 0.77 | 0.1 |
| HadGEM2-ESM | 0.83 | 0.89 | 0.06 | 0.89 | 0.93 | 0.04 |
| MIROC-ESM | 0.72 | 0.72 | 0.0 | 0.92 | 0.96 | 0.04 |

**Tab.8: Mean forest biome fraction in Northern Eurasia (0-150°E, 60-80°N) in the PFT-based biomisiations and the climate-based biomisations for the mid-Holocene (6k) and pre-industrial (PI) time-slice, and the difference between both (6k-PI).**

70



| | PI | | LGM | |
|---|---|---|---|---|
| | **FPC-method** | **PFT-method** | **FPC-method** | **PFT-method** |
| FSS (vs RF99) | 0.1 | 0.13 | - | - |
| κ (vs RF99) | 0.59 | 0.63 | - | - |
| κ (vs. records) | 0.19 | 0.24 | 0.17 | 0.13 |
| BNS (vs. records) | 0.53 | 0.55 | 0.54 | 0.41 |


**Tab.9: Metrics quantifying the agreement of the biomisations for IPSL-ESM-T63 based on the FPC-method (Prentice et al. 2011) or the PFT-based method introduced in this study with the modern potential biome distribution according to Ramakutty and Foley (1999, RF99) or pollen-based biome reconstructions (BIOME6000 database) for the pre-industrial (PI) and the Last Glacial Maximum (LGM) time-slices. Listed are the relative fractional skill score (FSS), the kappa value (κ) and the best neighbour score**

**(BNS).**



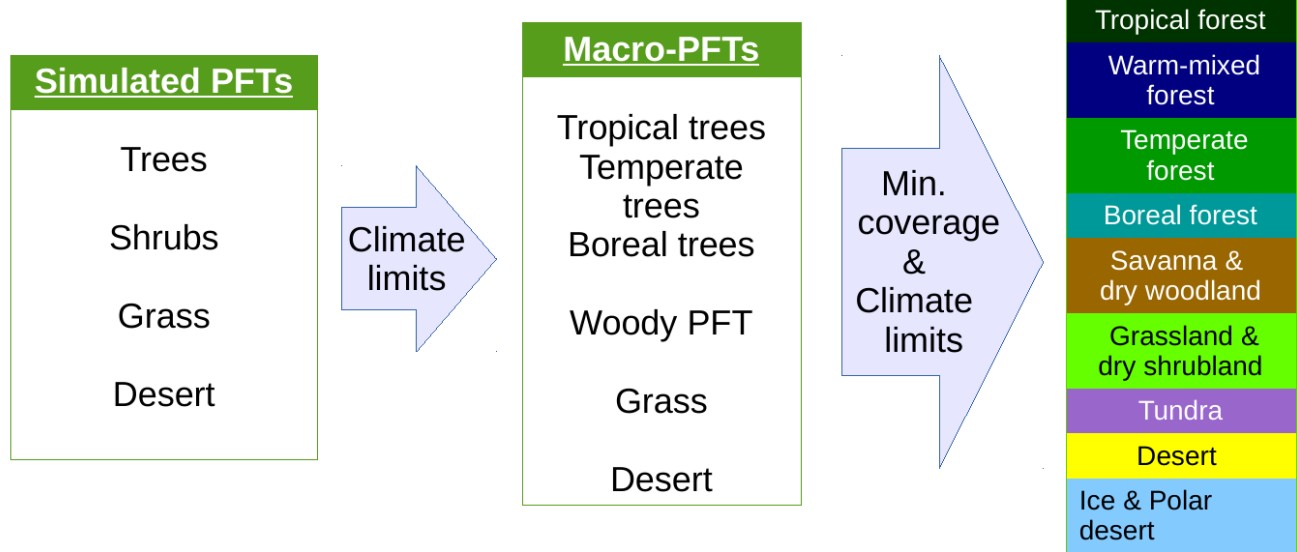

**Figure 1: Scheme of the biomisation method. The plant functional type (PFT) fractions simulated by the individual dynamic global vegetation models are assigned to Macro-PFT groups by climate limitation rules and are afterwards assigned to nine mega-biomes by assumptions on the minimum coverage of certain Macro-PFTs needed in a grid-cell and additional climate limits.**







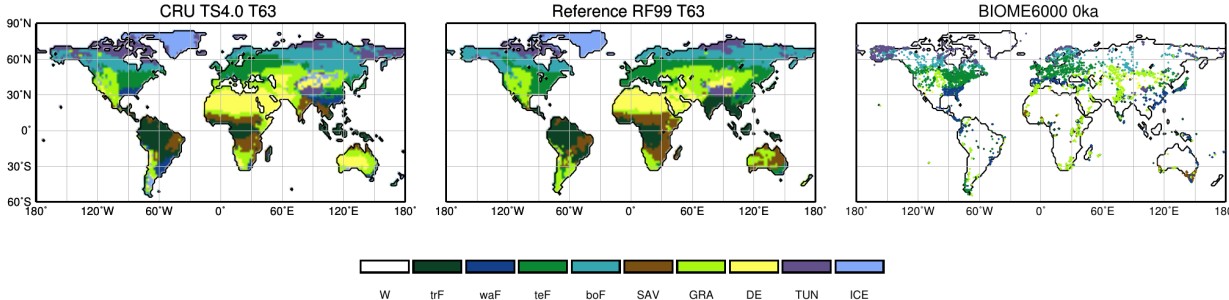

**Figure 2: Reference biome distributions for the pre-industrial time-slice, i.e. left panel: the biome distribution inferred by BIOME1 that has been forced by the CRU TS 4.0 dataset (period 1901-1930), interpolated to a Gaussian T63 grid; central panel: the modern potential natural vegetation map derived by Ramankutty and Foley (1999, RF99) remapped on a T63 gaussian grid; right panel: pollen-based pre-industrial biome reconstructions provided by the Biome6000 database (Harrison, 2017). The biomes are tropical forest (trF), warm-mixed forest (waF), temperate forest (teF), boreal forest (boF), Savanna & dry woodland (SAV), grassland & dry shrubland (GRA), warm desert (DE), Tundra (TUN) and polar desert & Ice (ICE).**





**Figure 3: Simulated pre-industrial mega-biome distributions according to the new biomisation method (PFT-based method). The plant functional type (PFT) fractions simulated by the individual models have been converted into mega-biomes through climate limitation rules and assumptions on the maximum coverage of certain PFTs needed in the grid-cells.**





**Figure 4: Simulated pre-industrial biome distributions according to the classical biomisation approach, i.e. biomising the climate states simulated by the individual models. The climate field were used to force the biome model BIOME1 (Prentice et al. 1992). Afterwards, the original BIOME1 biomes havebeen aggregated into the nine mega-biomes used in this study.**





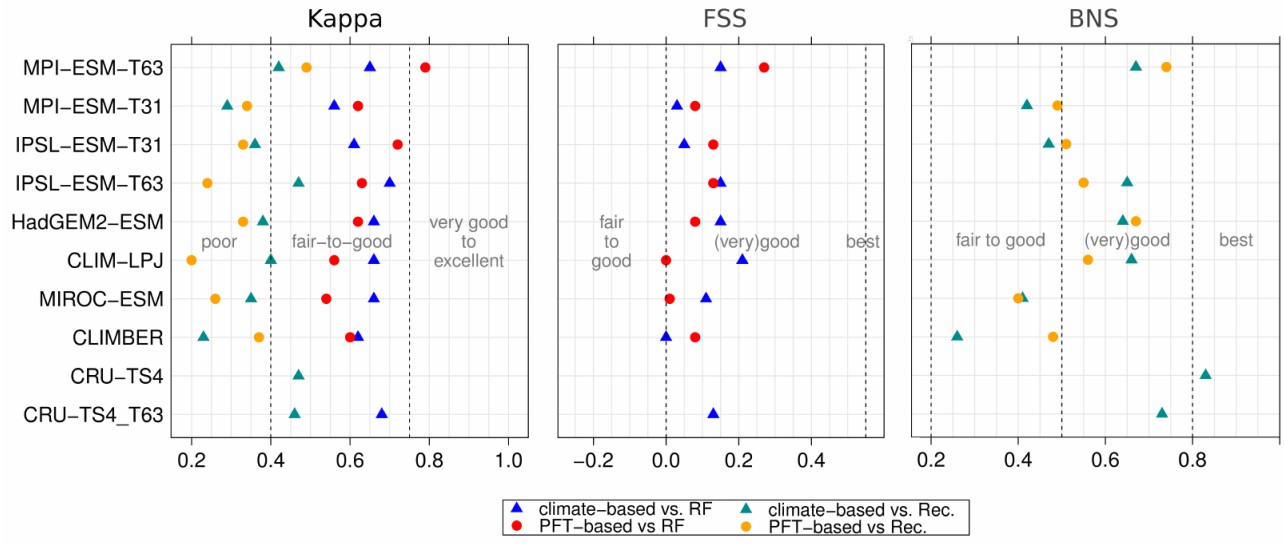

**Figure 5: Metrics quantifying the total agreement of the simulated pre-industrial biome maps based on the PFT cover fractions (PFT-method) or based on the climate state (classical approach using BIOME1) with the reference datasets (i.e. modern potential natural vegetation (RF) and pre-industrial pollen-based biome reconstructions (Rec.)). Shown are the Kappa values (left figure), the relative fractional skill score (FSS, middle figure) and the best neighbour score (BNS, right figure) for all models and also for the biomisation based on the CRU-TS4 observational climate data in original resolution (CRU-TS4) and interpolated to a T63 grid (CRU-TS4_T63).**





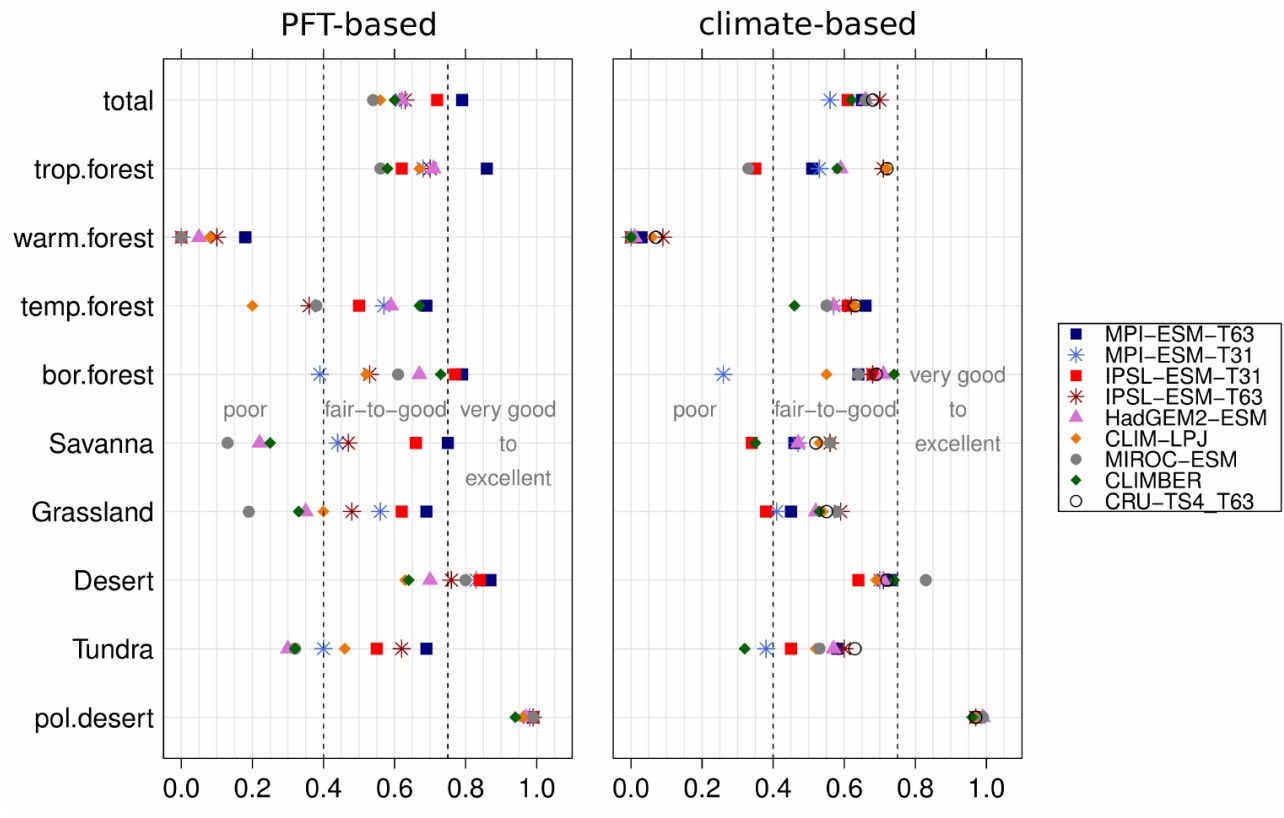

**Figure 6: Kappa metric quantifying the agreement of the simulated pre-industrial individual mega-biomes with the reference dataset (i.e. modern potential natural vegetation (RF)) for the PFT-based method (left panel) and the classical method using BIOME1 forced with the simulated background climate states (right panel).**






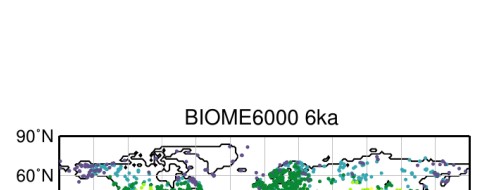

**Figure 7: Simulated mid-Holocene biome distribution in the different models, based on the PFT-method (left panel), pollen-based reconstructions of the mid-Holocene biome distribution (BIOME6000 database, upper figure) and the best neighbour score (BNS) for all individual sites showing the agreement of the reconstructed biomes and the biome distribution in the neighbourhood of the sites, ranging from 0 (no grid-cell in the surrounding shows the same biome as reconstructed) to 1 (the grid-cell locating the site and the record at the site indicate the same biome).**






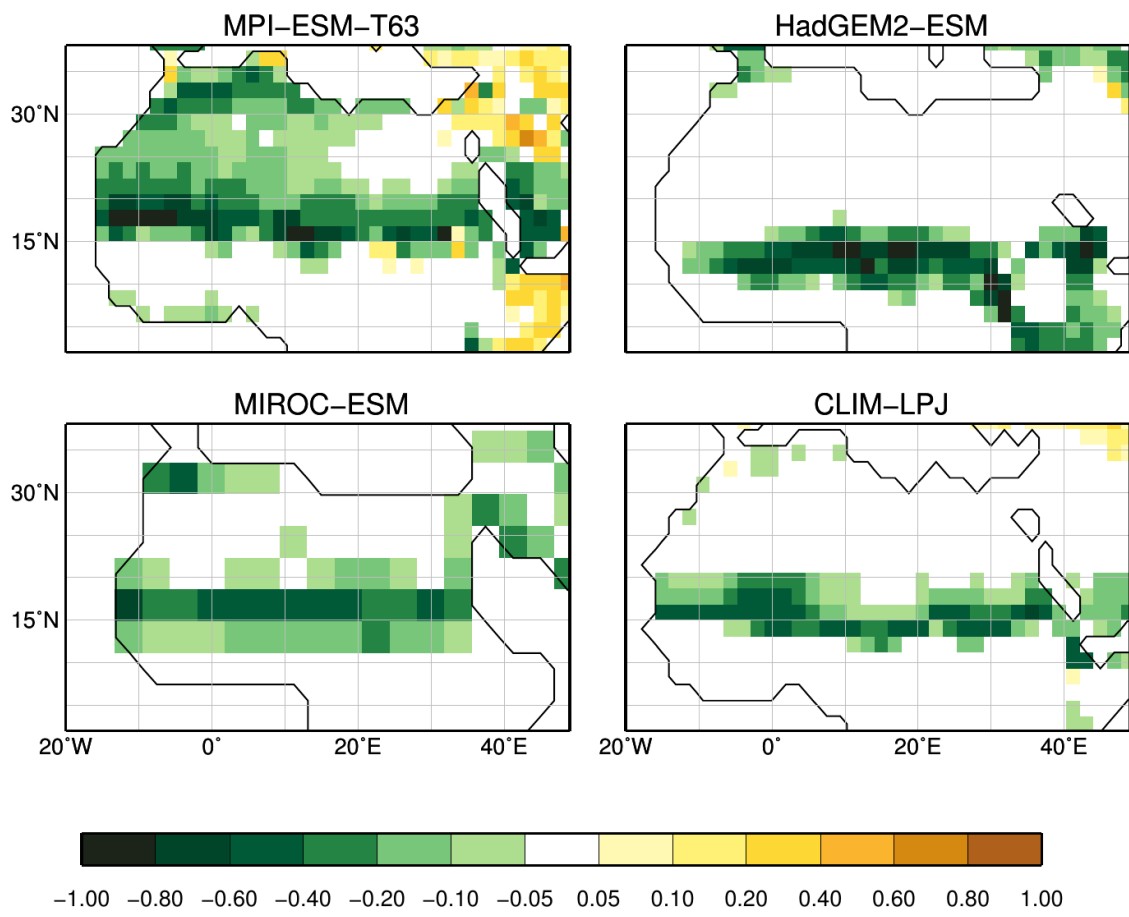

**Figure 8: Differences in the desert fractional coverage simulated by the individual models between the mid-Holocene (6k) and pre-industrial time-slice (0k).**




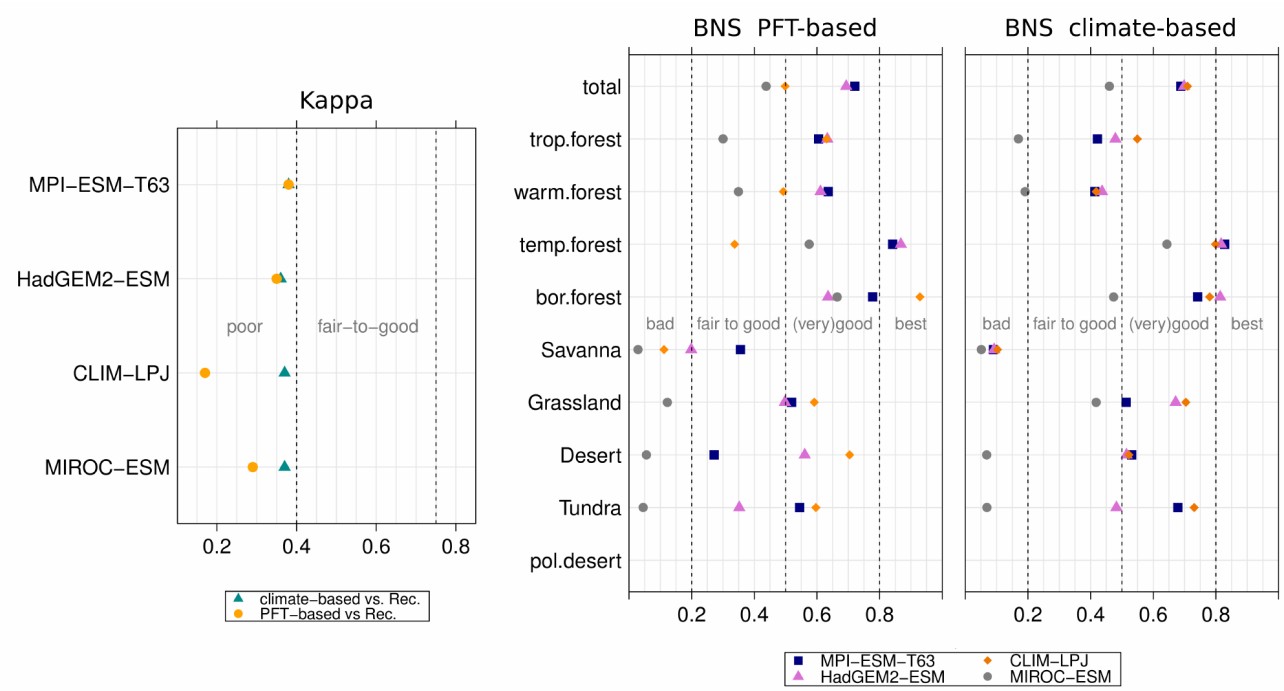

**Figure 9: Metrics quantifying the agreement of the simulated mid-Holocene biome maps based on the PFT-method or based on the climate states (i.e. according to the BIOME1 model)with the pollen-based biome reconstructions (BIOME6000 database) for the mid-Holocene time-slice, i.e. the (total) Kappa value (left panel) and the the BNS values for the individual mega-biomes.**


90



**Figure 10: Pollen-based biome reconstructions (BIOME6000 database) for the Last Glacial Maximum time-slice and the simulated biome distributions according to the new biomisation method (i.e. the PFT-based method)**



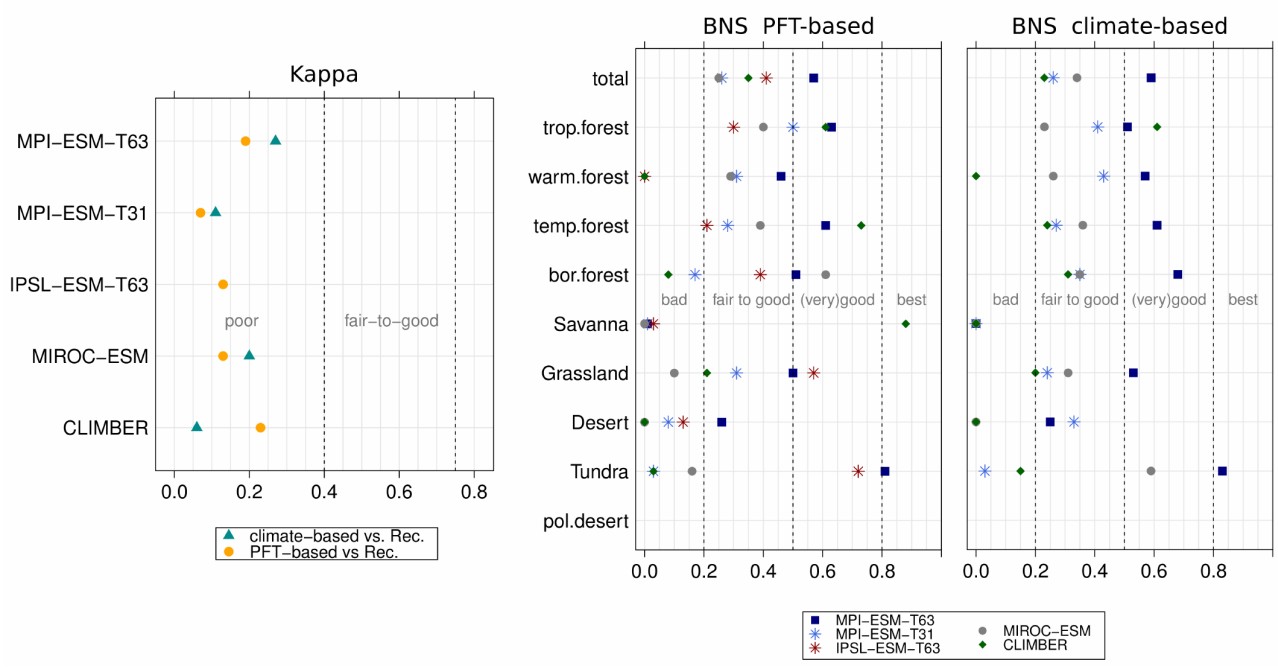

**Figure 11: Metrics quantifying the agreement of the simulated Last Glacial Maximum biome maps based on the PFT-method or based on the simulated climate states (i.e. according to BIOME1) with pollen-based biome reconstructions (BIOME6000 database), i.e. the (total) Kappa value and the BNS values for the individual mega-biomes. Please notice that the climatic variables needed to force BIOME1 could not be provided for IPSL-ESM-T63. Thus, no climate-based biomisation exists for IPSL-ESM-T63.**





**Figure 12: Climate factors leading to the differences between the pre-industrial climate-based biome distributions and the biome distribution inferred from the CRU-TS4.0 observational climate data. The factors were calculated by performing a sensitivity test with the BIOME1 model following Dallmeyer et al. (2017)**





**Figure 13: Zonal sum of pre-industrial forest biome area per latitude [Mio km²/degree] in the reference (RF99), the climate-based biomisation (blue) and the PFT-based biomisation (red) for each of the individual models. Due to the special land-sea mask in CLIMBER, the values for this model have been scaled by a factor of 0.766, which is the quotient of the global land area in a Gaussian T31 grid and the global land area in the CLIMBER grid (10°x10°).**





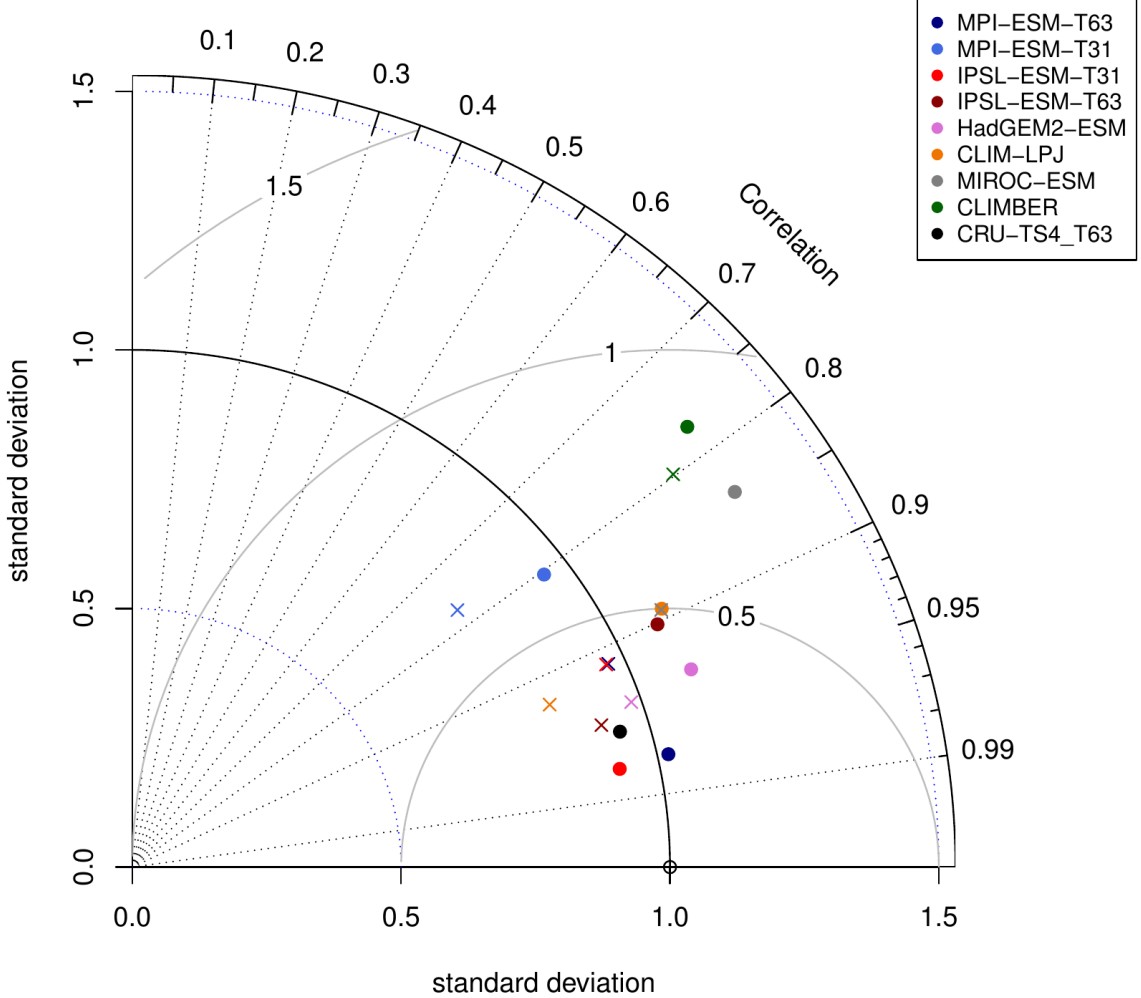

**Figure 14: Normalised Taylor-diagramm showing the agreement of the simulated pre-industrial zonal sum of forest biome area per latitude using BIOME1 (i.e. based on the simulated climate states, crosses) or the PFT-based method (dots) for the individual models with the modern potential biome distributions according to Ramakutty and Foley (1999, RF99). Additionally shown is the agreement of the CRU-TS4-based biomisation with this RF99 reference dataset.**





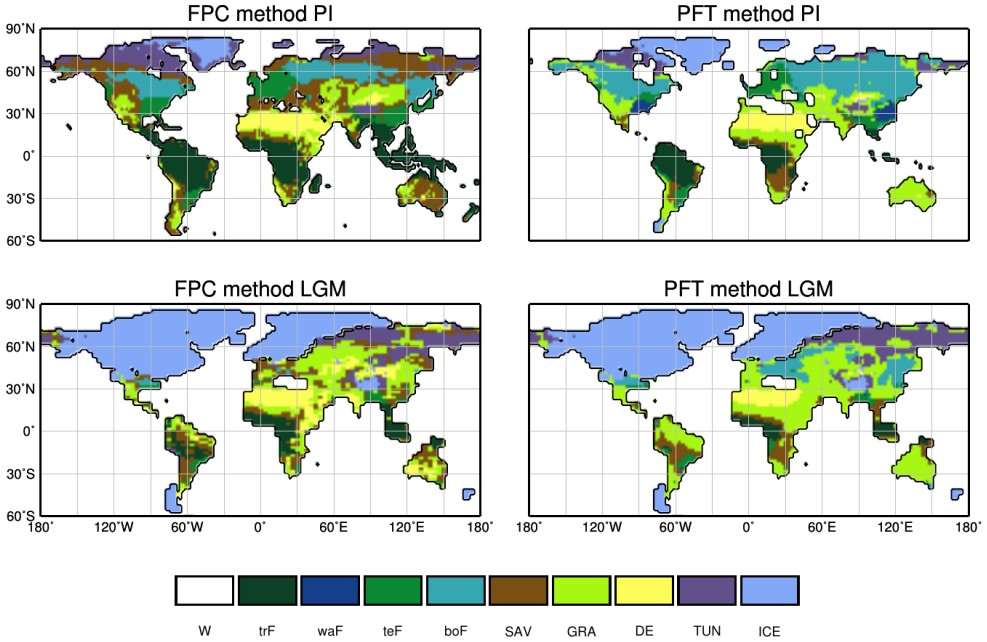

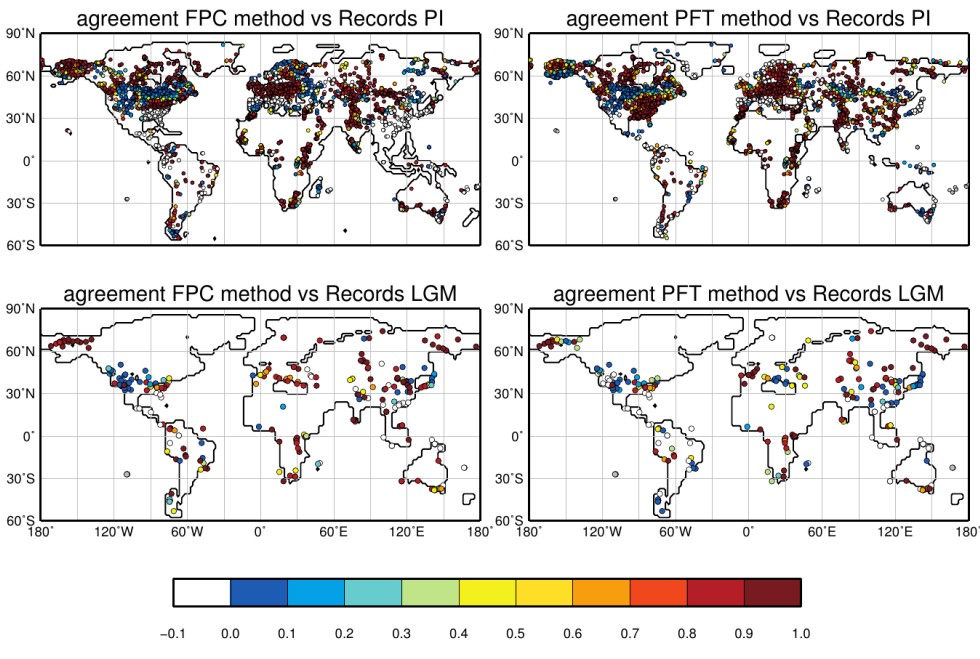

**Figure 15: Comparison between the biomisation method by Prentice et al. (2011, referred to as FPC-method) and the method introduced in this study (PFT-based) based on the model IPSL-ESM-T63. Shown are the biome distributions for the pre-industrial (PI) and Last Glacial Maximum (LGM) time-slice (upper panels) and the BNS for all available sites showing the agreement of the reconstructed biomes (according to the BIOME6000 database) and the simulated biome distributions in the neighbourhood of the sites (lower panels), ranging from 0 (no grid-cell in the surrounding shows the same biome as reconstructed) to 1 (the grid-cell locating the site and the record at the site indicate the same biome).**