# Peer review of "Harmonizing plant functional type distributions for evaluating Earth System Models"

_Climate of the Past, 2018_

## Referee Comment (RC1) · Anonymous Referee #1 · 22 Jun 2018

This manuscript describes and evaluates a new method for converting the output of earth system models (ESMs) into a description of vegetation cover that may be used to evaluate the performance of individual models, and to quantitatively compare a result among models that integrates several aspects of the climate system into a single variable. Vegetation model output has been used as a quick evaluation tool of climate and fully coupled earth system model output for decades, as the global distribution of vegetation is influenced by a range of climate parameters, e.g. temperature and precipitation, and the seasonal characteristics of climate. A global map of biomes produced with climate model output is readily qualitatively compared to observations, e.g., in the form of a map of potential natural vegetation, and may also be directly compared with paleoecological reconstructions, without the need, for example, of converting pollen assemblages, which reflect vegetation cover, to an estimate of climate conditions, which requires a transfer function or inverse modeling approach. As the authors note, most modern earth system models contain a dynamic, i.e., fully coupled, representation of vegetation cover. This provides the opportunity of using the ESM output of vegetation cover directly rather than translating modeled climate into vegetation cover using an offline vegetation model (e.g., the widely applied BIOME4 equilibrium global vegetation model).

Generally, this manuscript is well written, if rather long and tedious to read, and it describes a thorough study that will make a valuable contribution to the field. Ultimately, the scheme applied in the manuscript may become a standard methodology for intercomparison among ESMs, and for evaluation in the light of observations and paleoecological reconstructions. However, given that the authors emphasize that the manuscript describes a method rather than focusing on results, arguably the manuscript is not really suitable for *Climate of the Past* and should be published in a methods journal such as *Geoscientific Model Development*. The method described here would certainly not be limited to paleo-environmental applications, and could be widely applied, e.g., to model simulations for future scenarios.

My next major issue with this study is that authors appear to have overlooked some key literature and methodology on comparison of modeled vegetation with reconstructions. In particular the authors should have acknowledged and used the Delta-V method (Sykes et al., 1999) for comparing vegetation maps rather than the Kappa statistic. Delta-V is a robust method specifically designed for evaluating biome maps produced by vegetation models, that accounts for similarity between biome types in a way that Kappa cannot. If the authors feel that Delta-V is not applicable to their study, they should explain this in the manuscript. Along with using Delta-V, I have a number of specific comments on the text, which I describe below. Overall, this manuscript should be acceptable for publication after moderate revision.

Specific comments:

**Line 28** Change to "...these thresholds represent..."

**Line 67** In the mention of FPC here, and as discussed elsewhere in the manuscript, it is not clear here how this quantity differs from PFT cover fraction. Sitch et al. (2003) explicitly equate gridcell-level FPC with PFT cover fraction (pg. 165, eqn. 8 and following discussion). In my understanding of the dynamic vegetation schemes of the ESMs used in this study, all would take a similar approach of equating FPC with cover fraction. Some more explanation, either here or at another point in the manuscript would be helpful.

**Line 68** Here and at many other locations in the manuscript, the biome name "savanna" should not be capitalized. While many researchers mistakenly equate the biome "savanna" with the person or place name "Savannah", purposes of this manuscript savanna is not a proper name, and neither it nor any other biome names should be capitalized.

**Line 94-95** I understand the PFT group names to represent super-groupings of PFTs. In this case, I can accept PFT groups called desert, forest, and grass, but the name "wood" does not make sense – how is this supposed to be different from the forest group. Some more clarification and precision of the terminology would be helpful here.

**Line 107-109** The bioclimatic limits listed here needs citations to justify the choices made, or further explanation of how these values and thresholds were determined.

**Line 111** Again, where does the arbitrary choice of a 2°C threshold in mean annual temperature for warm to cold desert come from? The widely used Köppen-Geiger classification separates hot deserts from cold deserts at a mean annual temperature threshold of 18°C (e.g., Kottek et al., 2006, and references therein). Provide citations or empirical justification for your choice of threshold.

**Line 113** Change "residuum" to "residual"

**Line 151** It appears some text is missing after the word "following.."

**Line 154-155** The sentence "SEIB is a gap model..." needs further explanation to describe how it is different from other dynamic vegetation schemes. LPJ-GUESS is also a gap model, and it is also used in ESMs. It is not clear to the reader what is meant by this sentence and how it affects the results of the study.

**Line 169-170** Given the authors emphasis "that this study is an introduction of a new biomisation method and not an evaluation of the different vegetation models..." it would have been helpful to focus the manuscript more on the methods and less on the very long evaluation sections that follow. In addition to Figure 1, I would have liked to see a flowchart showing the detailed decisionmaking process for classifying PFTs into PFT groups (or macro-PFTs as they are termed in the figure but not in the text) and then into mega-biomes, where all of the thresholds and other classification parameters are also included.

**Line 179** The sentences starting with "As temperature threshold..." is not sufficiently linked to the previous lines – It is not clear what temperature threshold you are talking about. Clarify.

**Line 183-184** Upon what basis or citation is the decision to set the limit for existence of C4 grass at a MTCM of 10°C? C4 grasses grow in climates with much colder winters in North America and Eurasia (see e.g., Still et al., 2003).

**Line 186** How is the "dominant mega-biome type" selected. What if there is a tie among types? What if no type constitutes a majority of the cover (e.g., if there were three or more biomes present)?

**Line 210** Please repeat the exercise using Delta-V (Sykes et al., 1999), or explain why you opted to not use this statistic, which is arguably more appropriate than Kappa when comparing vegetation maps. The FSS method described in section 2.4.2 appears to

be designed to deal with spatial offset, but not similarity or difference of vegetation categories.

**Line 255** While the biomised pollen data the authors used in this study (BIOME6000) does not contain metadata about pollen catchment size, so it is impossible to know over what surface the BIOME6000 points are integrating. Nevertheless, in this section the spatial implication of the best neighbor score should be quantified, i.e., the comparison between model and biomised pollen data could be influenced by points how many km distant from either the pollen site or the nearest gridcell center?

**Line 279-280** It is not clear from the world map figures – these are reproduced at a size that is much to small to discern anything but the broadest patterns – how the the "area covered by tropical forests is... more in line with observations". To clarify this point, provide a detail map to (as in figure 8) and/or summary statistics, such as total area covered by the biome in the simulations vs. the natural vegetation map.

**Line 288-289** At the spatial resolutions used in this study, it is extremely difficult to resolve anything at the scale of Alaska (and impossible to see given the small size of the figures). Both BIOME1 and the vegetation models do not faithfully reproduce the vegetation of Alaska, so the meaningfulness of this comparison and the assertion in these lines is questionable.

**Line 311** Why didn't the authors aggregate the biomised pollen sites on to the same grid used by the ESMs, similar to the process used by Trondman et al. (2015). Some areas have relatively few point observations of pollen spectra, while others many sites. A modal gridding procedure could have provided a more fair comparison between the models and the pollen-based reconstructions.

**Line 353** It seems that Figure 8 should be called out at this point in the manuscript but it is not. The maps in Figure 7 are not sufficiently large to appreciate what is written in the text. Also the phrase "According to the records,..." needs a citation. What records are these?

**Line 377** The phrase "e.g. Europe was mostly covered by grassy biomes..." is incorrect. It appears from the figure that, and in reality, Europe was mostly covered by ice. The assertion that Europe was largely treeless during the LGM is controversial and most previous modeling work has been inconclusive (Prentice et al., 2011). The most recent pollen-based reconstructions leave open the possibility of substantial woodland cover in parts of LGM Europe (Kaplan et al., 2016). A way to solve this for the current manuscript would be to delete this example from the text and choose another, less controversial example.

**Line 401** The assertion that the "biome 'warm-mixed forest' (subtropical forest) ... shares most tree species with tropical evergreen broadleaf forests" cannot be true, the citation to Ni et al., 2010 notwithstanding. First of all, the very high floristic diversity in the tropics and subtropics means that almost certainly different species grow in different habitats along a temperature gradient from the equator to the mid-latitudes. Perhaps some genera are commonly found in both biomes, but even then, most taxa are different, and perhaps the only thing these two biomes have in common is the presence of broadleaf evergreen trees, although even then, warm temperate broadleaf evergreen species are physiologically very different from true tropical species (e.g., Walter et al., 1971). Warm mixed forests are further distinguished from tropical forests by the presence of gymnosperms, which are rare to nonexistent in warm tropical forests. These characteristics distinguishing the biomes are acknowledged in the original BIOME1 paper (Prentice et al., 1992).

**Line 403** The citation to Chen et al., 2010 is not included in the bibliography, but I am anyway skeptical about this statement, because as noted above, the presence of conifer taxa in an assemblage of otherwise (sub-)tropical pollen spectrum are indicative of warm mixed forest conditions, which is precisely how this biome can be identified (Prentice et al., 2000). More explanation/clarification are necessary here.

**Line 408** The statement that "...warm-mixed forest is based on temperate broadleaved trees..." is incorrect, as this biome also, critically, includes needlelaf trees, which distinguishes it from, e.g., tropical forests. Even the BIOME1 paper recognizes this. The sentence should be clarified.

**Line 415** While it is described later in the manuscript that this study only considers tropical savannas, the statement "Savannas [sic] require the coexistence of trees and C4-grass" is incorrect and must be clarified at this point in the text.

**Line 423** Indeed, it is difficult to reconstruct tropical savanna on the basis of pollen data because tropical trees produce relatively little windborne pollen. A citation here would also be helpful, (e.g., Jones et al., 2011).

**Line 467-468** It is not clear how biomised model output (biomes) is compared to the CRU-TS4 dataset (climate). I am missing a step in here. What was compared precisely? Clarify.

**Table 1** This table appears to contain some of the information concerning the aggregation of model PFTs into PFT groups, but this is not indicated in the first column of the table, nor is the terminology consistent with what is used in the manuscript text and in Figure 1. Where is the PFT group called "wood" in the manuscript text?

**Table 2** Provide a reference or empirical rationale for choosing these limits.

**Table 3** Provide references or empirical rationale for the choice of parameters and thresholds in this table.

**Figure 1** Some further explanation is needed to explain how the "simulated PFTs" relates to the long list of PFTs listed in Table 1. Also It is not clear how trees get classified into either a "... trees" (or "forest" in the text) PFT group vs. a "Woody PFT" (or "wood" group in the text). Clarification and careful standardization of the terminology across the text, tables, and figures, is essential for any revision of this manuscript.

**Figures 2-4, 7, 10, 15** The maps presented here are generally two small for the reader to make anything but the most superficial of comparisons. Especially Figure 2 is too small, but the maps should be enlarged in all of the figures as much as possible and

presented in a standard size, e.g., by providing letter codes for each map within the map frame and then maximizing the size of the maps by removing the labels and whitespace around each map. Also the latitude and longitude labels are not necessary in this type of presentation and can be removed. To the contrary, the color legend should not use abbreviated labels, but instead give the full name of each mega-biome, so that readers can quickly interpret the figure without having to return to the text.

**References**

Jones, H. T., Mayle, F. E., Pennington, R. T., and Killeen, T. J.: Characterisation of Bolivian savanna ecosystems by their modern pollen rain and implications for fossil pollen records, Rev Palaeobot Palyno, 164, 223-237, 2011. https://doi.org/https://doi.org/10.1016/j.revpalbo.2011.01.001.

Kaplan, J. O., Pfeiffer, M., Kolen, J. C. A., and Davis, B. A. S.: Large Scale Anthropogenic Reduction of Forest Cover in Last Glacial Maximum Europe, PLoS One, 11, e0166726, 2016. https://doi.org/10.1371/journal.pone.0166726.

Kottek, M., Grieser, J., Beck, C., Rudolf, B., and Rubel, F.: World Map of the Koeppen-Geiger climate classification updated, Meteorologische Zeitschrift, 15, 259-263, 2006. https://doi.org/10.1127/0941-2948/2006/0130.

Prentice, I. C., Cramer, W., Harrison, S. P., Leemans, R., Monserud, R. A., and Solomon, A. M.: A Global Biome Model Based on Plant Physiology and Dominance, Soil Properties and Climate, J Biogeogr, 19, 117-134, 1992. https://doi.org/10.2307/2845499.

Prentice, I. C., Harrison, S. P., and Bartlein, P. J.: Global vegetation and terrestrial carbon cycle changes after the last ice age, New Phytol, 189, 988-998, 2011. https://doi.org/10.1111/j.1469-8137.2010.03620.x.

Prentice, I. C., Jolly, D., and BIOME 6000 participants: Mid-Holocene and glacial-maximum vegetation geography of the northern continents and Africa, J Biogeogr, 27, 507-519, 2000. https://doi.org/10.1046/j.1365-2699.2000.00425.x.

Sitch, S., Smith, B., Prentice, I. C., Arneth, A., Bondeau, A., Cramer, W., Kaplan, J. O., Levis, S., Lucht, W., Sykes, M. T., Thonicke, K., and Venevsky, S.: Evaluation of ecosystem dynamics, plant geography and terrestrial carbon cycling in the LPJ dynamic global vegetation model, Global Change Biol, 9, 161-185, 2003. https://doi.org/10.1046/j.1365-2486.2003.00569.x.

Still, C. J., Berry, J. A., Collatz, G. J., and DeFries, R. S.: Global distribution of C3 and C4 vegetation: Carbon cycle implications, Global Biogeochem Cy, 17, 6-1-6-14, 2003. https://doi.org/10.1029/2001gb001807.

Sykes, M. T., Prentice, I. C., and Laarif, F.: Quantifying the impact of global climate change on potential natural vegetation, Climatic Change, 41, 37-52, 1999. https://doi.org/Doi 10.1023/A:1005435831549.

Trondman, A. K., Gaillard, M. J., Mazier, F., Sugita, S., Fyfe, R., Nielsen, A. B., Twiddle, C., Barratt, P., Birks, H. J. B., Bjune, A. E., Bjorkman, L., Brostrom, A., Caseldine, C., David, R., Dodson, J., Dorfler, W., Fischer, E., van Geel, B., Giesecke, T., Hultberg, T., Kalnina, L., Kangur, M., van der Knaap, P., Koff, T., Kunes, P., Lageras, P., Latalowa, M., Lechterbeck, J., Leroyer, C., Leydet, M., Lindbladh, M., Marquer, L., Mitchell, F. J.

G., Odgaard, B. V., Peglar, S. M., Persson, T., Poska, A., Rosch, M., Seppa, H., Veski, S., and Wick, L.: Pollen-based quantitative reconstructions of Holocene regional vegetation cover (plant-functional types and land-cover types) in Europe suitable for climate modelling, Global Change Biol, 21, 676-697, 2015. https://doi.org/10.1111/gcb.12737.

Walter, H., Mueller-Dombois, D., and Burnett, J. H.: Ecology of Tropical and Subtropical Vegetation, Oliver  Boyd, Edinburgh, 1971.

---

## Referee Comment (RC2) · Anonymous Referee #2 · 20 Aug 2018

**General comments**

Dallmeyer et al present a new method to convert plant functional type (PFT) distributions simulated by earth system models into biomes ("biomisation") that can be more readily compared with other earth system and biome models (of varying complexity and number of PFTs) and with paleo-reconstructions of past vegetation distributions that are typically classified at the biome level. The authors highlight the difficulties of classifying plant species into wider groups, and the challenges posed when comparing different types of classifications (such as PFTs and biomes). However, I have two main concerns with the study that could potentially be alleviated with greater clarity in the text and/or with some restructuring and reframing of the paper. These are detailed more below and in the specific comments section.

[Figure]

The introduction explains clearly why a model-generic PFT-based biomisation is needed, as opposed to other model-specific methods that have been used in the past. However, it is not obvious to me from the text why this new method that the authors are proposing is needed beyond the "classical approach" they compare to, and therefore how much this is study is a novel contribution to the literature. All methods rely on subjective decisions as to how to classify vegetation into biomes, therefore adding another may only contribute to uncertainty in this area. This is particularly the case given that both the PFT-based and classical methods compare reasonably well; therefore, I am left wondering why the classical approach cannot be used and why a new method is needed? As the authors state in lines 338-340 "In general, the skill in representing the individual mega-biomes is similar for the PFT- and the climate-based method. Both approaches have the same strengths and weaknesses, but the spread between the models is larger for the PFT-based biomisations." And in lines 346-347 "Overall, the metrics indicate that the PFT-based method works similarly well as the classical approach of biomising climate states." What then is the specific value of this new PFT-based biomisation method? I perhaps have misunderstood (but therefore likely some other readers will too) – so more clarity is needed on this in the introduction. Biomisation itself should be defined in the introduction, and the classical approach needs to be defined there, and not at the end of Section 2.1, and a detailed explanation should be provided as to why we cannot use this method and instead need the new PFT-based method.

It is clear the authors have thought in depth about performing a rigorous analysis and the caveats of their analysis are well detailed in the discussion. The analysis of the differences between models and comparison of different mega-biomes (as shown in Fig. 6 for example) is interesting and useful for the modeling community, particularly given the authors are testing whether the number of PFTs, model complexity, grid cell resolution, or simulation land use contribute to these differences. However, this is not a stated objective or important outcome of the paper. I understand the authors do not want to focus on a model comparison because the same climate data were not used to

drive the models, and therefore the authors have chosen to focus on the introduction of the new PFT-based biomisation method. However, I am not sure this is a strong enough focus to sell the paper given the PFT-based and classical biomization methods produce similar results, as I discussed above. Furthermore, much of the text reads as a model comparison, thus causing confusion as to the objectives of the paper. And the authors claim this is a new powerful method that is needed in order to more readily compare models (as well as models with reconstructions), but then say this study cannot be a model comparison because they were not forced with the same climate states. It seems perhaps redundant to introduce a new method for a particular purpose and then not use it for that purpose. Clearly there is a lot of useful analysis and discussion in this paper, but I think it would be more useful if it were to be re-framed and re-structured with different objectives.

Finally, the manuscript is somewhat dense in places, with many figures to digest, likely owing to the difficulty of describing complex and detailed analyses. The reader would likely benefit if the authors could spend some time editing the text to make it more concise.

Specific comments

Introduction Lines 45-50: It might be useful if the authors detail the differences between diagnostic biome models forced with GCM-derived climate data vs DGVMs coupled to GCMs? Similarly, it might be useful to detail the difference between PFT and biome classifications. Examples could help with this. And finally, it might be useful in line 50 to clarify again that simulations from the DGVMs have been disregarded because PFT classifications are different to biome classifications.

Methods

Section 2.1 Tables 2 and 3 and for all bioclimatic thresholds (lines 99-111): Is there a reference for the bioclimatic limits used here? Why have these bioclimatic limits been used? Is there evidence to suggest these are better descriptors of bioclimatic limits

than used in other studies?

Lines 115-119: Please could the authors give more detail here? Perhaps they could provide more detail as to what they mean by "biomise the underlying climate". An illustrative example would be helpful.

Section 2.2 It would be interesting to denote which simulations in Table 4 are run with dynamic vs static vegetation. Lines 154-156: Please could the authors briefly explain what is the difference between a gap model and a tiling approach. This will aid a reader who is less familiar with these models. Further, please could they describe how the PFT distribution has been determined via the NPP of the vegetation categories for SEIB and why the vegetation categories do not already correspond to the PFT distribution?

Section 2.3 Line 175: Please could the authors briefly explain how Haxeltine and Prentice (1996) derived their vegetation compilation and what it includes?

Line 180: I do not understand this "on a basis of 5°C being higher than 900°C derived from modern observations". Please could the authors explain what they mean here? Does this correspond with Table 3? If so, it might be useful to say that.

Line 203: "biomisation of simulated climate states (i.e. the classical method)". This is the first time the classical method has been referenced. It might be useful to the reader to describe it earlier and to detail how the authors' method differs from this classical method – particularly as it is used to compare the new method with throughout the results section. Otherwise, it appears somewhat out of the blue here.

Results

Section 3.1 It might be more constructive to compare the PFT- and classical climate-based biomisation methods for each model side by side in one figure, and not the former in Figure 3 and latter in Figure 4, given the authors compare the two. It is hard to compare each model given they are on separate pages.

Line 287: It would be instructive to reference that "better represented" means in comparison to observations that are provided in Figure 2.

Discussion

Section 4.1 is well described and an honest and comprehensive view of the caveats of the methods.

Technical corrections Ni et al. (2010) missing most of the reference Dallmeyer et al (2017) reference missing → check all references in the bibliography Line 97: additionally → additional Line 147: has → have Line 148: what is the importance of knowing that the simulations have been re-done on a new computer? Will they be better as a result? Line 159: extra → specifically Line 168: "than the other simulations..." It would be helpful to add "that were run with dynamic/interactive vegetation". Line 187: details → detail Line 399: works → work Line 420: "very rudimentary" → "in a very rudimentary manner" Line 521: coverage → cover

---

## Referee Comment (RC3) · Anonymous Referee #3 · 22 Aug 2018

As requested , I have reviewed the article entitled "Harmonizing plant functional type distributions for evaluating Earth System Models" by Anne Dallmeyer, Martin Claussen and Victor Brovkin for "Climate of the Past".

In this study, the authors have developed a new method for the biomisation of simulated plant functional type (PFTs) distributions with different global vegetation models. They have tested this method for the pre-industrial, mid-Holocene and Last Glacial Maximum time-slices. This approach, named "PFT-based biomisation" is compared with biome distributions inferred from the classical approach based on the biome model BIOME1 (Prentice, 1992), and with pollen-inferred biomes (0 ka, 6 ka and LGM) and estimates of the potential natural vegetation (0 ka). They evidenced some mismatches between simulated and reconstructed biomes but the results also suggest that the method works

well for all models and is comparable to the other biomisation techniques used here (Prentice et al. 1992, 2011).

I think that the paper of Dallmeyer et al. is a serious work and presents interesting findings in terms of method and results. The choice of Climate of the Past is, for me, appropriate: even if this paper mainly focus on vegetation models, it is also based on general circulation models. I think this paper is clear, well written and can be published in Climate of the Past with minor changes, which are listed below:

- abstract: must be reworked, it should be more informative on the methods, the results and more precise (method, key results, conclusions) : how many models are used, which is the principe of the method. . .

- introduction: this point concerns the originality of this paper, you need to better justified the objective and the questions of your paper given that at least two biomisation methods exist and work well (Prentice et al., 1992, 2011). The new biomisation method must be more clearly defined.? What is "natural" PFTS (line 36)? Line 44, the ref Prentice et al., (1996) is needed.

-methods -biomisation: line 91: what is mean growing degree days? GDD0 or 5? Please define it. Could you also define the "multi-year mean PFT cover fractions". Why don't you use the alpha parameter classically used in Prentice et al (1992, 1996). Line 109: GDD0 exceeds 800°C or to the biome 'tundra', if GDD0 is below 800°C: could you explain the choice of these values? -simulations: line 121 "Simulations from nearly all state-of-the-art global dynamic vegetation models": the exact number of models is needed here as the name of the models. "Overall, eight simulations for the pre-industrial climate (PI) and vegetation, four for mid-Holocene (6k) conditions and five for Last Glacial Maximum (LGM) conditions have been used (Tab.4)": if I look at tab4, I find 6 simulations for PI, 3 for 6 ka, and 4 for LGM: please check; the format of tab 4 is not easy to read: correct it. Line 140: what is CRUNCEP? -preparing the reference datasets: OK for the 6 ka and LGM but I have a problem with the comparison for the PI

[Figure]

Period. The comparison with the Preindustrial changes is an important point to validate the results. However, the models uses the pre-industrial period as a baseline whereas the pollen-inferred biomes use the late 20th century (modern pollen data used for the 0 ka in biome 6000 have been collected from 1960 to present day, so they don't correspond to preindustrial period). On the same way, in the RF99 dataset, Ramankutty and Foley used a global representation of permanent croplands in 1992, which not represent the PI period. This points of the discussion must be added and discussed in depth ($CO_2$ bias and human impact).

-results: -lines 350-3522: you forget to discuss the increase of the temperate forest in europe at 6 ka

-discussion -caveats in the method: line 401: the biome warm mixed forest is not only subtropical but also appears in Europe: please correct. Line 432: could you define the "anthropogenic plant functional types"? line 446: you state that the procedure of reconstructing paleovegetation often include modern analogue technique. I don't agree with that. The MAT is used to reconstruct the climate, not the biomes. The biome procedure using pollen follows the biomisation defined by prentice et al 1996, and peyron et al 1998. . .using a pollen-taxa –PFT assignment and a PFT-biome assignment which is done on modern samples (0 ka), and fossil samples (6 and LGM) independently. Please correct the text. - biases in the PI. . ., line 455: the classical method of biomising.. a ref is needed here. line 468: could you give more details on the sensitivity study performed by Dallmeyer et al 2017?line 479: NPP ratios, not the acronym. line 483: what do you mean by low score? Its not clear on the fig.

- figures: too many figures! I strongly recommend to group the figures 2 and 3, and also the figures 9 and 11 ; the figures 4 and 8 can be moved to suppl. Material (not discussed in depth and not very useful).

-references : please check carefully your references, some are missing (Dallmeyer et al., 2017; Tian et al., 2017. . .) or not homogenized.

other points: -in the text, you often write "pollen-based reconstructions" or "reconstructions"; I don't like this terminology because the pollen also is used to reconstruct climate or other : could you replace in the text by pollen-based (or inferred) biomes which is more precise and used. -homogenize in the text: pi, PI, Pi -figures: -fig.1: Could you replace on the fig. "climate limits" by "bioclimatic limitation" and refer in the caption to tab.1 (climate limits) and tab 3 (Min cov. and Climate limits); the 4 simulated PFTs doesn't correspond to the text (p3: desert, forest, wood, grass and total vegetation): please check! -fig.2 and others: could you change the color of grass? The green one is not very clear and can be confused with the temperate forest, orange will be better and commonly accepted by the biomisation community. This fig. must be group with fig 3. -fig.12: Not clear: what is the range of the climate factor? Could you explain better how they are calculated and not just refer to Dallmeyer et ay 2017? -fig.15: even if it's explained in the text, its strange to see the biome savanna in Europe and Medit. Area (may be explain better in the fig. caption).

---

## Author Comment (AC1) · 6 Nov 2018

We would like to thank the anonymous Referee #1 for her/his constructive comments which helped to significantly improve the manuscript.

R: Generally, this manuscript is well written, if rather long and tedious to read, and it describes a thorough study that will make a valuable contribution to the field. Ultimately, the scheme applied in the manuscript may become a standard methodology for intercomparison among ESMs, and for evaluation in the light of observations and paleoecological reconstructions. However, given that the authors emphasize that the manuscript describes a method rather than focusing on results, arguably the manuscript is not really suitable for Climate of the Past and should be published in a methods journal such

as Geoscientific Model Development. The method described here would certainly not be limited to paleo-environmental applications, and could be widely applied, e.g., to model simulations for future scenarios.

Our response: The major aim of our study is twofold: a) introduction of a new PFT-based biomisation method for comparing different Earth System Model simulations and b) application of the method to two cases, mid-Holocene and LGM. Perhaps our method is also applicable to simulations of future climate, but we do not have tested it. Therefore, we have chosen submission of our paper to Climate of the Past. Referee 3 explicitly agrees with our choice. However, we leave it to the decision of the editor whether the revised manuscript would be better published in GMD.

R: My next major issue with this study is that authors appear to have overlooked some key literature and methodology on comparison of modelled vegetation with recon-structions. In particular the authors should have acknowledged and used the Delta-V method (Sykes et al., 1999) for comparing vegetation maps rather than the Kappa statistic. Delta-V is a robust method specifically designed for evaluating biome maps produced by vegetation models that accounts for similarity between biome types in a way that Kappa cannot. If the authors feel that Delta-V is not applicable to their study, they should explain this in the manuscript. Along with using Delta-V, I have a number of specific comments on the text, which I describe below.

Our response: We fully agree with the referee that the Delta-V method is a power-ful statistic to quantify the difference between simulated biome distributions. Though, Delta-V was designed to quantify the strength of change between biome maps by con-sidering and weighting changes in attributes and plant forms and considering the dom-inance of the plants in the biomes. Biome models such as BIOME4 or BIOME1 include many biomes that differ only slightly in the PFT composition. In our study, we use mega-biomes that (mostly) differ strongly from each other. Changes in attributes such as the phenology of the plants do not play a role for this biomisation method, reduc-ing the advantages of the Delta-V method. In addition, we do not focus on the biome

changes between the time-slices, but on the differences between the simulated biome maps at a certain time. We assume that the maps do not deviate strongly from each other and from the references. As Sykes et al. (1999) state, the "kappa statistic is well suited to assessing the degree of similarity between two maps that are expected a priori to agree". The further advantage of the kappa statistic is that it is a very widespread, well-known method. Thus, we think the kappa statistic is an appropriate method for this study, though, we are aware of the weaknesses of this statistic (e.g. treating all biome differences equivalently). The other methods (FSS and BNS) are very easy to understand and apply and are able to cope with slight pattern displacements. We will consider including the Delta-V statistic in the biomisation tool for upcoming research.

Specific comments:

R: Line 28 Change to " . . . these thresholds represent ..." Our response: done.

R: Line 67 In the mention of FPC here, and as discussed elsewhere in the manuscript, it is not clear here how this quantity differs from PFT cover fraction. Sitch et al. (2003) explicitly equate grid cell-level FPC with PFT cover fraction (pg. 165, eqn. 8 and following discussion). In my understanding of the dynamic vegetation schemes of the ESMs used in this study, all would take a similar approach of equating FPC with cover fraction. Some more explanation, either here or at another point in the manuscript would be helpful.

Our response: Even if the FPC could be calculated for all models, the vegetation height is not a standard output in the CMIP5 database. So a comparison of the method based on all models is not possible. We decided to move the comparison between the PFT-method and the method by Prentice et al. to the Appendix E as the comparison is only based on one model and as the inclusion of temperate savanna in the biome distribution provided by Zhu et al. hampers the comparability with the PFT-biomisations.

R: Line 68 Here and at many other locations in the manuscript, the biome name "savanna" should not be capitalized. While many researchers mistakenly equate

the biome "savanna" with the person or place name "Savannah", purposes of this manuscript savanna is not a proper name, and neither it nor any other biome names should be capitalized.

Our response: We carefully looked through the manuscript again and corrected all capitalized biome names and change 'savannah' to 'savanna'.

R: Line 94-95 I understand the PFT group names to represent super-groupings of PFTs. In this case, I can accept PFT groups called desert, forest, and grass, but the name "wood" does not make sense – how is this supposed to be different from the forest group. Some more clarification and precision of the terminology would be helpful here.

Our response: We agree, this is misleading. 'Wood' includes all shrubs and forest PFTs. We add this information and change the names from forest to trees and from wood to woody PFTs. We now write: "In detail, the PFTs calculated by the respective dynamic vegetation model are aggregated into the groups 'trees', 'woody PFTs' (i.e. shrubs and all tree PFTs), 'grass' and 'desert' which is calculated by one minus the total vegetation" and changed the PFT group names in the following text consistently.

R: Line 107-109 The bioclimatic limits listed here needs citations to justify the choices made, or further explanation of how these values and thresholds were determined.

Our response: The bioclimatic limits (with few exceptions) follow the limitations defined in the biome model BIOME4, we mentioned this in the lines 89-91 in the originally manuscript, but we agree that we have not emphasized this enough. We deleted Tab.2 and include detailed information on the references in the caption of Tab. 3 (now Tab.2). We add to the caption of Tab.3 (now Tab.2): …. "Bioclimatic limits and assumptions on minimum PFT coverage needed for the assignment of PFTs into the PFT-groups and into the 9 Mega-Biomes. The separation of boreal, temperate and tropical tree-PFTs is based on the same bioclimatic limits as the respective forest mega-biomes. Only temperature-based limitations are used, i.e. the growing degree days on a basis of 5°C

(GDD5) or on a basis of 0°C (GDD0), the monthly mean temperature of the coldest month (Tc), and the annual mean temperature (Tann). Bioclimatic limits are mainly taken from the BIOME4 model (Kaplan et al., 2003, marked with * ). The limit for tropical forest is taken from BIOME1 (Prentice et al., 1992), but is also commonly used in DGVMs (e.g. JSBACH). The limit for the differentiation of deserts has been empirically determined in this study and is close to the value chosen by Handiani et al. (2013) and within the range of the Koeppen-Geiger climate classification for polar climate and the Holdridge alpine life zone classification. The Tc limit for warm savannas is taken from JSBACH (C4-grass criteria) to exclude temperate savannas. The assumptions on minimum coverage have partly been taken, partly been empirically adapted from Handiani et al. (2013)..."

R: Line 111 Again, where does the arbitrary choice of a 2°C threshold in mean annual temperature for warm to cold desert come from? The widely used Köppen-Geiger classification separates hot deserts from cold deserts at a mean annual temperature threshold of 18°C (e.g., Kottek et al., 2006, and references therein). Provide citations or empirical justification for your choice of threshold.

Our response: We agree that the information on the biomisation criteria were imprecise. We tested several criteria for the differentiation of warm to cold desert (e.g. the BIOME1 criteria of 22°C for the temperature of the warmest month), but all criteria fail to separate the polar from the temperate and hot desert. Even the criteria chosen in Handiani et al. (2013) of a mean annual temperature above 0°C for the (warm) desert lead to grid cells with polar deserts in Asia (Taklamakan and Gobi) in some models. We decided to choose 2°C, because it was the lowest temperature (closed to the limit of Handiani et al.) that lead to a reasonable distribution of the polar desert. In addition, the limit is well in the range of the Koeppen-Geiger climate classification of polar climate (tundra climate: temperature of the warmest month between 0-10°C, ice cap climate: temperature during all months below 0°C) and the limit (Tann < 3°C) of the Holdridge life zone classification). In Tab.3 (now Tab.2) we wrote 'cold desert' which

was misleading. We changed it to 'polar desert'. Furthermore, we add information on the biomisation criteria in the table caption (see last comment).

R: Line 113 Change "residuum" to "residual" Our response: done.

R: Line 151 It appears some text is missing after the word "following.." Our response: We corrected it.

R: Line 154-155 The sentence "SEIB is a gap model ..." needs further explanation to describe how it is different from other dynamic vegetation schemes. LPJ-GUESS is also a gap model, and it is also used in ESMs. It is not clear to the reader what is meant by this sentence and how it affects the results of the study.

Our response: We further explained the difference between SEIB and the other models in the revised manuscript: "SEIB deviates from the other DGVMs in this study as it does not use the tiling approach of calculating PFT fractional coverage for each grid-cell. It is a so called forest gap-model, simulating the interactions among individual trees that compete for light and space in arising gaps (e.g. due to disturbances) within a spatially explicit virtual forest. The model was built for capturing the vegetation dynamics on local scale. The application of the model for larger (e.g. global) scales is possible, but global simulations partly disagreed with observations (Sato et al., 2007). The PFT distribution used in this study has been calculated in the post-processing for CMIP5 via the relative net primary productivity of the vegetation categories, it was not explicitly calculated by the model, which may lead to additional biases in the vegetation distribution."

R: Line 169-170 Given the authors emphasis "that this study is an introduction of a new biomisation method and not an evaluation of the different vegetation models ..." it would have been helpful to focus the manuscript more on the methods and less on the very long evaluation sections that follow.

Our response: We agree that the evaluation of the biomisation method is long and we

slightly shorten it in the revised manuscript. We furthermore move the comparison with the FPC method to the Appendix. Our intention for the study was the implementation of a tool that works for present day but also palaeo-timeslices and can be applied for all DGVMs. Therefore, we decided to include not only the mid-Holocene, but also the LGM time-slice, enlarging the study.

R: In addition to Figure 1, I would have liked to see a flowchart showing the detailed decisionmaking process for classifying PFTs into PFT groups (or macro-PFTs as they are termed in the figure but not in the text) and then into mega-biomes, where all of the thresholds and other classification parameters are also included.

Our response: This is indeed a very helpful suggestion, we included such a flowchart in the Appendix for the VECODE model, i.e. the model with the fewest PFTs. Unfortunately this flowchart looks different for the other models, because all models use different PFTs. We will include detailed information on the biomisation for all models (e.g. flowchart) in the README file of the biomisation Tool, which will be provided when this manuscript will be finally published.

R: Line 179 The sentences starting with "As temperature threshold ..." is not sufficiently linked to the previous lines – It is not clear what temperature threshold you are talking about. Clarify.

Our response: We revised this sentence: "RF99 additionally includes the biome 'Evergreen/Deciduous Mixed Forest/Woodland'. Here, this biome is assigned to the mega-biomes 'temperate forest' in warm regions and 'boreal forest' in colder regions via the modern growing degree days distribution ($GDD5 \geq 900°C$ for warm region, $GDD5 < 900°C$ for cold region), derived from observations (University of East Anglia Climatic Research Unit Time Series 3.1, University of East Anglia, 2008, Harris et al., 2012)."

R: Line 183-184 Upon what basis or citation is the decision to set the limit for existence of C4 grass at a MTCM of 10°C? C4 grasses grow in climates with much colder winters

in North America and Eurasia (see e.g., Still et al., 2003).

Our response: Thank you for this information, we adopt this limit from the bioclimate criteria for C4 grass in the JSBACH model and it worked well. The limit MTCM of 17°C taken in other biomisation approaches (e.g. Roche et al., 2007 or Handiani et al. 2013) was too high.

R: Line 186 How is the "dominant mega-biome type" selected. What if there is a tie among types? What if no type constitutes a majority of the cover (e.g., if there were three or more biomes present)?

Our response: We calculate the dominant mega-biome in the model grid-cell by counting the number of grid-cells covered by this biome in the RF99 original grid. We explained it in the text: "Within each of this model grid cells, the dominant mega-biome type in the 5-minute-resolved RF99 data was taken for covering the RF99 grid-box in T31 or T63 or in the 10°grid. In more details, each grid-box on a T31 Gaussian grid contains 45*45 grid-cells of the 5-minute-resolved RF99 data. Within these 45*45 grid-boxes the fractional coverage of all mega-biomes is calculated and the biome with the highest fraction is chosen for covering the T31 grid-box. ….." If there is a tie among types, this is not considered. In lots of grid-cell three or even more biomes are present, but in all grid-cells only one biome is dominant, which we take as mega-biome for this cell.

R: Line 210 Please repeat the exercise using Delta-V (Sykes et al., 1999), or explain why you opted to not use this statistic, which is arguably more appropriate than Kappa when comparing vegetation maps. The FSS method described in section 2.4.2 appears to be designed to deal with spatial offset, but not similarity or difference of vegetation categories.

Our response: Please see the comment above, for this study, we decided not to use the Delta-V statistic.

R: Line 255 While the biomised pollen data the authors used in this study (BIOME6000) does not contain metadata about pollen catchment size, so it is impossible to know over what surface the BIOME6000 points are integrating. Nevertheless, in this section the spatial implication of the best neighbor score should be quantified, i.e., the comparison between model and biomised pollen data could be influenced by points how many km distant from either the pollen site or the nearest gridcell center?

Our response: This is indeed problematic, but so far, the BIOME6000 database is the best data available. As mentioned in the original text, the size of the neighbourhoods depends on the grid of the model data. For T31 we choose three grid-boxes in each direction, i.e. 11.25° on a Gaussian grid in each direction, for T63 we choose six grid-cells in each direction (i.e. 11.25° on a Gaussian grid) and for the 10°grid, we choose only one grid-cell in each direction. These regions seems to be very large in 'reality', but not in the model world. Due to the weighting by a distance decay function, the furthest off grid-cells of the neigborhood have only a very minor influence on the score. We included some examples on the meaning of the BNS, translated into distances: "For instance, a BNS of approx. 0.82 or 0.46 means that (in the mean) the best neighbour grid-cell is among the grid-cell 'circle' next to the site-locating grid-cell in T63 or T31, respectively. Accordingly, a BNS of 0.04 indicates a distance between the best neighbour and the site-locating grid-cell of of 7.5° on a Gaussian grid."

R: Line 279-280 It is not clear from the world map figures – these are reproduced at a size that is much to small to discern anything but the broadest patterns – how the the "area covered by tropical forests is . . . more in line with observations". To clarify this point, provide a detail map to (as in figure 8) and/or summary statistics, such as total area covered by the biome in the simulations vs. the natural vegetation map.

Our response: Indeed, the main aim of these global figures is to capture the main differences in the broad pattern (just by eye). A detailed validation of the simulated biome pattern is given by the metrics. As South America is an 'outstanding' region, because the tropical forest biome is strongly underestimate in the climate based biomisations

(for most models) and well captured in most of the PFT-based biomisations, we include a Table in the Appendix to verify this statement.

R: Line 288-289 At the spatial resolutions used in this study, it is extremely difficult to resolve anything at the scale of Alaska (and impossible to see given the small size of the figures). Both BIOME1 and the vegetation models do not faithfully reproduce the vegetation of Alaska, so the meaningfulness of this comparison and the assertion in these lines is questionable.

Our response: Thank you very much for carefully reading the manuscript. Alaska is indeed imprecise. We mean Alaska and the north-western parts of Canada. We corrected this in the revised version.

R: Line 311 Why didn't the authors aggregate the biomised pollen sites on to the same grid used by the ESMs, similar to the process used by Trondman et al. (2015). Some areas have relatively few point observations of pollen spectra, while others many sites. A modal gridding procedure could have provided a more fair comparison between the models and the pollen-based reconstructions.

Our response: The gridding of the biomised pollen sites is much effort and is not the subject of this study. We are no experts for pollen-based reconstructions. We just used the best global synthesis available for us to evaluate our model results.

R: Line 353 It seems that Figure 8 should be called out at this point in the manuscript but it is not. The maps in Figure 7 are not sufficiently large to appreciate what is written in the text. Also the phrase "According to the records, ..." needs a citation. What records are these?

Our response: We refer to the BIOME6000 reconstructions, shown in Fig.7. The records are not shown in Fig.8. We shift the '(Fig.7)' -reference to the end of this sentence and inserted 'BIOME6000'. Again, the aim of the figure is giving a broad (not detailed) overview of the biome distributions in the different simulations. We did not

enlarge the figures to make possible that they still can be plotted on one page.

R: Line 377 The phrase "e.g. Europe was mostly covered by grassy biomes … " is incorrect. It appears from the figure that, and in reality, Europe was mostly covered by ice. The assertion that Europe was largely treeless during the LGM is controversial and most previous modeling work has been inconclusive (Prentice et al., 2011). The most recent pollen-based reconstructions leave open the possibility of substantial woodland cover in parts of LGM Europe (Kaplan et al., 2016). A way to solve this for the current manuscript would be to delete this example from the text and choose another, less controversial example.

Our response: We agree with the referee that this is not the best example and deleted this sentence in the manuscript.

R: Line 401 The assertion that the "biome 'warm-mixed forest' (subtropical forest) … shares most tree species with tropical evergreen broadleaf forests" cannot be true, the citation to Ni et al., 2010 notwithstanding. First of all, the very high floristic diversity in the tropics and subtropics means that almost certainly different species grow in different habitats along a temperature gradient from the equator to the mid-latitudes. Perhaps some genera are commonly found in both biomes, but even then, most taxa are different, and perhaps the only thing these two biomes have in common is the presence of broadleaf evergreen trees, although even then, warm temperate broadleaf evergreen species are physiologically very different from true tropical species (e.g., Walter et al., 1971). Warm mixed forests are further distinguished from tropical forests by the presence of gymnosperms, which are rare to nonexistent in warm tropical forests. These characteristics distinguishing the biomes are acknowledged in the original BIOME1 paper (Prentice et al., 1992). and R: Line 403 The citation to Chen et al., 2010 is not included in the bibliography, but I am anyway skeptical about this statement, because as noted above, the presence of conifer taxa in an assemblage of otherwise (sub-)tropical pollen spectrum are indicative of warm mixed forest conditions, which is precisely how this biome can be identified (Prentice et al., 2000). More explanation/clarification are

necessary here.

Our response: We apologize for misinterpreting the literature and the discussion with palynologists. We discussed it again and corrected this sentence to: "The mega-biome 'warm-temperate forest' (e.g. subtropical forest) includes PFTs that can be assigned to several biomes and is rather defined by a co-existence of certain PFTs. For instance, in the BIOME4 model it is not only defined by the dominance of temperate evergreen broadleaved trees, but can also be defined by a dominance of cool conifers (with sub-PFT of temperate evergreen broadleaved trees). The cool conifers – in turn - are also part of temperate forest biomes. Given the limited number of PFTs in the DGVMs , the confinement of biomes via PFT-mixtures is not possible. As biome models such as BIOME4 generally manage to simulate warm-temperate forest at the correct locations, we adopt the bioclimatic limits from BIOME4 (limit for temperate evergreen broadleaved trees) for defining this mega-biome. Nevertheless, the calculated warm-temperate forest distribution strongly disagree with the reference datasets. The warm-temperate forest shares most subtropical tree species with the tropical evergreen forest (Ni et al, 2010). These biomes tend to overlap in some regions and are sometimes mixed up in reconstructions (Chen et al, 2010). In addition, this mega-biome includes the 'warm temperate rainforest' and the 'wet sclerophyll forest and woodland' in the BIOME6000 reconstructions (cf. Harrison, 2017), which may not be able to be identified with our biomisation method. Regarding the modern reference of RF99, we decided to assigned the biome 'temperate needle-leaf evergreen forest and woodland' of the RF99 dataset to the mega-biome 'temperate forest', although, this biome is also located e.g. in the southern USA which should be assigned to the warm-temperate forest. Therefore, the evaluation of this method with respect to warm-temperate forest might be ambiguous. The correct reproduction of the warm-temperate forest distributions further hampered by the rather regional distribution of the warm-temperate forest covering only few grid-cells." We added the reference of Chen et al to the references: Chen, Y., Ni, J., Herzschuh, U., 2010.: Quantifying modern biomes based on surface pollen data in China. Glob. Planet. Chang. 74, 114-131. R: Line 408 The statement that "...

warm-mixed forest is based on temperate broadleaved trees ..." is incorrect, as this biome also, critically, includes needlelaf trees, which distinguishes it from, e.g., tropical forests. Even the BIOME1 paper recognizes this. The sentence should be clarified.

Our response: We changed this sentence to (see above): "As biome models such as BIOME4 generally manage to simulate warm-temperate forest at the correct locations, we adopt the bioclimatic limits from BIOME4 (limit for temperate evergreen broadleaved trees) for defining this mega-biome."

R: Line 415 While it is described later in the manuscript that this study only considers tropical savannas, the statement "Savannas [sic] require the coexistence of trees and C4-grass" is incorrect and must be clarified at this point in the text.

Our response: We add the word 'tropical' : "While tropical savannas require the co-existence of trees and C4-grass, they can only be distinguished from forests by their unique functional ecology, fire tolerance and shade intolerance (Ratman et al., 2011)"

R: Line 423 Indeed, it is difficult to reconstruct tropical savanna on the basis of pollen data because tropical trees produce relatively little windborne pollen. A citation here would also be helpful, (e.g., Jones et al., 2011).

Our response: Thank you very much for bringing this problem to our mind. We included it in the revised manuscript: "Modern pollen rain analysis reveal that woody plant taxa typically characterizing savannas are under-represented or even absent in the pollen/vegetation ratios. Given the lack of savanna indicators, this biome may be overlooked in fossil pollen records (Jones et al. 2011)."

R: Line 467-468 It is not clear how biomised model output (biomes) is compared to the CRU-TS4 dataset (climate). I am missing a step in here. What was compared precisely? Clarify. Our response: We agree that is indeed not precisely described. We now write: " ...we compare the pre-industrial climate-based biomisations with the biomisation of the CRU-TS4 dataset. ..."

[Figure]

R: Table 1 This table appears to contain some of the information concerning the aggregation of model PFTs into PFT groups, but this is not indicated in the first column of the table, nor is the terminology consistent with what is used in the manuscript text and in Figure 1. Where is the PFT group called "wood" in the manuscript text?

Our response: This table has nothing to do with the aggregation of the PFTs into the PFT groups used in the biomisation. We just divided the PFTs in different PFT classes to structure this table. We added the header 'PFT class' to the first column.

R: Table 2 Provide a reference or empirical rationale for choosing these limits. and R: Table 3 Provide references or empirical rationale for the choice of parameters and thresholds in this table.

Our response: We provided detailed references on the bioclimatic limits and the thresholds in the table caption (see comment above). Please notice that we deleted Tab.2.

R: Figure 1 Some further explanation is needed to explain how the "simulated PFTs" relates to the long list of PFTs listed in Table 1. Also It is not clear how trees get classified into either a "... trees" (or "forest" in the text) PFT group vs. a "Woody PFT" (or "wood" group in the text). Clarification and careful standardization of the terminology across the text, tables, and figures, is essential for any revision of this manuscript.

Our response: This is indeed misleading. We changed the caption to: "...The plant functional type (PFT) fractions simulated by the individual dynamic global vegetation models (DGVM) are assigned to the PFT groups 'desert' (i.e. 1-total vegetation), 'grass' (containing all grass PFT-types) 'woody PFT' (containing all trees and shrub types) and 'trees' (containing all tree types). The 'trees' and 'woody PFTs' are further differentiated into 'tropical trees', 'temperate trees', 'temperate woody PFTs' and 'boreal woody PFTs' via bioclimatic limitations (Tab.2). For DGVMs explicitly distinguishing tropical, temperate or boreal tree types, the original classification of the DGVMs is used. Afterwards the PFT-groups are assigned to nine mega-biomes by assumptions on the minimum coverage of certain PFT-groups needed in a grid-cell and additional bioclimatic

limitations (Tab.2)." We furthermore changed the column header and the terminology across the text etc. accordingly.

R: Figures 2-4, 7, 10, 15 The maps presented here are generally two small for the reader to make anything but the most superficial of comparisons. Especially Figure 2 is too small, but the maps should be enlarged in all of the figures as much as possible and presented in a standard size, e.g., by providing letter codes for each map within the map frame and then maximizing the size of the maps by removing the labels and whitespace around each map. Also the latitude and longitude labels are not necessary in this type of presentation and can be removed. To the contrary, the color legend should not use abbreviated labels, but instead give the full name of each mega-biome, so that readers can quickly interpret the figure without having to return to the text.

Our response: Thank you very much for these suggestions to improve the figures. We enlarge the figures by deleting the latitude and longitude labels and also change the colour legend. We decided to keep the model labels and not to use letter codes. We think, the reader can then find a certain model more quickly.

---

## Author Comment (AC2) · 6 Nov 2018

R: The introduction explains clearly why a model-generic PFT-based biomisation is needed, as opposed to other model-specific methods that have been used in the past. However, it is not obvious to me from the text why this new method that the authors areproposing is needed beyond the "classical approach" they compare to, and therefore how much this is study is a novel contribution to the literature. All methods rely on subjective decisions as to how to classify vegetation into biomes, therefore adding another may only contribute to uncertainty in this area. This is particularly the case given that both the PFT-based and classical methods compare reasonably well; therefore, I am left wondering why the classical approach cannot be used and why a new method is needed? As the authors state in lines 338-340 "In general, the skill in represent-

ing the individual mega-biomes is similar for the PFT- and the climate-based method. Both approaches have the same strengths and weaknesses, but the spread between the models is larger for the PFT-based biomisations." And in lines 346-347 "Overall, the metrics indicate that the PFT-based method works similarly well as the classical approach of biomising climate states." What then is the specific value of this new PFT-based biomisation method? I perhaps have misunderstood (but therefore likely some other readers will too) – so more clarity is needed on this in the introduction.

Our response: We have to admit that the main aim of the new method obviously got lost in the introduction. The main point is that General Circulation Models (or Earth System Models) usually calculate not only the climate but also (if a DGVM is coupled in the General Circulation Model) a vegetation pattern that is in quasi-equilibrium with this climate. This vegetation is represented in form of plant functional type cover fractions, but these PFTs are not used in the 'classical' biomisation. The classical biome models (such as BIOME1) only use the simulated climate as forcing and calculates PFT distributions and then biome distributions on its own. Most DGVMs are more complex and include more relevant processes, therefore a more appropriate way would be to use these DGVM-simulated PFTs and not the climate pattern for the biomisation. And this is exactly what our method aims for. We introduce a method that can directly biomise the DGVM-simulated PFT distributions and this method performs similar well as the classical method (which is not necessarily the case as biome models are highly tuned) We further stress the problem of the classical method: "Using this conventional method of biomisation, fundamental palaeo-vegetation analysis can be undertaken (e.g. Jolly et. al. 1998; Harrison et al., 2003; Wohlfahrt et al., 2008; Harrison et al., 2016, Dallmeyer et al., 2017) without requiring an explicit calculated vegetation distribution by the Earth System Models. On the other hand this also means that existing simulated plant functional type distributions calculated by the DGVMs being dynamically coupled in these models are neglected, since only the simulated climate pattern is taken into account. The biomisation via diagnostic biome models did not include any information on the original PFT-distribution simulated by the Earth System

Models. As the DGVMs are generally more complex than the biome models and include more relevant processes, valuable information included in the PFT-distribution gets lost in the classical biomisation by the biome models. A more appropriate way of biomisation would be to use directly the PFT-distributions calculated by the DGVMs." and: "Therefore these methods can not directly be adopted for all existing dynamic vegetation models. A consistent inter-model comparison of the simulated vegetation distribution and an evaluation of the models against reconstructions on biome level is so far not possible. " and: "...we developed a biomisation technique that is based on the PFT-distributions simulated by the DGVMs and few input variables and simple differentiation rules...."

R: Biomisation itself should be defined in the introduction, and the classical approach needs to be defined there, and not at the end of Section 2.1, and a detailed explanation should be provided as to why we cannot use this method and instead need the new PFT-based method.

Our response: We agree that the introduction needs further clarifications. We have changed the paragraph to: "Pollen records are originally displayed in form of pollen percentages or pollen accumulation rates that can not be directly compared to plant functional type distributions, as pollen records do not reflect the actual plant abundances. For a systematic comparison of simulated plant functional type distributions and reconstructions, both need to be converted in a compatible format. In the last two decades, taxa to PFT assignment-methods and the method of 'biomisation' for pollen-based reconstructions have been developed (e.g. Prentice al. 1996, Ni et al., 2010, Harrison et al. 2010), so that pollen assemblages can be grouped into biomes (e.g. tropical forest, temperate steppe, desert). Pollen-based biome syntheses have been provided (Prentice et al., 1998 and 2000, Bigelow et al., 2003; Ni et al. 2010, Harrison et al., 2017, Tian et al. 2017) that have extensively been used to evaluate simulated biome distributions obtained from diagnostic biome models such as BIOME1 or BIOME4 (e.g. Prentice et al., 1992, Haxeltine and Prentice, 1996, Kaplan et al., 2003). These biome

models can be forced by observed or simulated climate fields and calculate biome distributions in equilibrium to this input climate. Using this classical method of biomisation, fundamental palaeo-vegetation analysis can be undertaken (e.g. Jolly et.al 1998; Harrison et al., 2003; Wohlfahrt et al., 2008; Harrison et al., 2016, Dallmeyer et al., 2017) without requiring an explicit calculated vegetation distribution in the General Circulation or Earth System Models. On the other hand this also means that existing simulated plant functional type distributions calculated by the DGVMs being dynamically coupled in these General Circulation or Earth System Models are neglected, since only the simulated climate pattern is taken into account. The biomisation via diagnostic biome models did not include any information on the original PFT-distribution simulated by the models. As the DGVMs are generally more complex than the biome models and include more relevant processes, valuable information included in the PFT-distribution gets lost in the classical biomisation by the biome models. A more appropriate way of biomisation would be to use directly the PFT-distributions calculated by the DGVMs."

R: It is clear the authors have thought in depth about performing a rigorous analysis and the caveats of their analysis are well detailed in the discussion. The analysis of the differences between models and comparison of different mega-biomes (as shown in Fig. 6 for example) is interesting and useful for the modeling community, particularly given the authors are testing whether the number of PFTs, model complexity, grid cell resolution, or simulation land use contribute to these differences. However, this is not a stated objective or important outcome of the paper. I understand the authors do not want to focus on a model comparison because the same climate data were not used to drive the models, and therefore the authors have chosen to focus on the introduction of the new PFT-based biomisation method. However, I am not sure this is a strong enough focus to sell the paper given the PFT-based and classical biomization methods produce similar results, as I discussed above.

Our response: In our revision we will more clearly outline how the new method differs from the classical biomisation method (i.e. the Biome Models) and why this new

method is useful (see last comments). That the methods produce similar results has to be seen as one of the strength of the new method, it can keep up with the tuned biome models.

R: Furthermore, much of the text reads as a model comparison, thus causing confusion as to the objectives of the paper. And the authors claim this is a new powerful method that is needed in order to more readily compare models (as well as models with reconstructions), but then say this study cannot be a model comparison because they were not forced with the same climate states. It seems perhaps redundant to introduce a new method for a particular purpose and then not use it for that purpose. Clearly there is a lot of useful analysis and discussion in this paper, but I think it would be more useful if it were to be re-framed and re-structured with different objectives.

Our response: We agree that this is confusing. Unfortunately, no simulation setup exists in which all different DGVMs were forced with the same climate state. We had to confine our study to the available simulations. The lack of the same background climate lead to differences between the biome distributions that are neither related to the method nor to the dynamic global vegetation model. Therefore, this study can only be an introduction and evaluation of the biomisation method based on different models, but we can not judge which dynamic vegetation model is the best. We carefully go through the text again to avoid phrases that read as a model comparison, e.g. we now rather use terms such as 'the biomisation of model X' to underline that we only compare different biomisations and not the models.

R: Finally, the manuscript is somewhat dense in places, with many figures to digest, likely owing to the difficulty of describing complex and detailed analyses. The reader would likely benefit if the authors could spend some time editing the text to make it more concise.

Our response: We shorten the text by moving parts of the informations on the simulations and the preparation of the reference datasets to the Appendix (including Tab.5).

We also move the comparison with the FPC-method to the Appendix. We furthermore deleted few sentences that were not so relevant. We deleted Tab.6 and Tab.2.

Specific comments

R: Lines 45-50: It might be useful if the authors detail the differences between diagnostic biome models forced with GCM-derived climate data vs DGVMs coupled to GCMs? Similarly, it might be useful to detail the difference between PFT and biome classifications. Examples could help with this. And finally, it might be useful in line 50 to clarify again that simulations from the DGVMs have been disregarded because PFT classifications are different to biome classifications.

Our response: We changed the text to: "In the last two decades, taxa to PFT assignment-methods and the method of 'biomisation' for pollen-based reconstructions have been developed (e.g. Prentice al. 1996, Ni et al., 2010, Harrison et al. 2010), so that pollen assemblages can be grouped into biomes (e.g. tropical forest, temperate steppe, desert). Pollen-based biome syntheses have been provided (Prentice et al., 1998 and 2000, Bigelow et al., 2003; Ni et al. 2010, Harrison et al., 2017, Tian et al. 2017) that have extensively been used to evaluate simulated biome distributions obtained from diagnostic biome models such as BIOME1 or BIOME4 (e.g. Prentice et al., 1992, Haxeltine and Prentice, 1996, Kaplan et al., 2003). These biome models can be forced by observed or simulated climate fields and calculate biome distributions in equilibrium to this input climate. Using this classical method of biomisation, fundamental palaeo-vegetation analysis can be undertaken (e.g. Jolly et. al., 1998; Harrison et al., 2003; Wohlfahrt et al., 2008; Harrison et al., 2016, Dallmeyer et al., 2017) without requiring an explicit calculated vegetation distribution in the General Circulation or Earth System Models. On the other hand this also means that existing simulated plant functional type distributions calculated by the DGVMs being dynamically coupled in these General Circulation or Earth System Models are neglected, since only the simulated climate pattern is taken into account. The biomisation via diagnostic biome models did not include any information on the original PFT-distribution simulated by the models.

As the DGVMs are generally more complex than the biome models and include more relevant processes, valuable information included in the PFT-distribution gets lost in the classical biomisiation by the biome models..."

Methods R: Section 2.1 Tables 2 and 3 and for all bioclimatic thresholds (lines 99-111): Is there a reference for the bioclimatic limits used here? Why have these bioclimatic limits been used? Is there evidence to suggest these are better descriptors of bioclimatic limits than used in other studies?

Our response: We added the references to the figure captions. Most bioclimatic limits were taken from the BIOME4 model.

R: Lines 115-119: Please could the authors give more detail here? Perhaps they could provide more detail as to what they mean by "biomise the underlying climate". An illustrative example would be helpful.

Our response: We now write: "To assess the performance of the biomisation based on simulated PFTs, we additionally biomise the simulated climate fields corresponding to the PFT-distributions for each model. This is the conventionally used procedure to biomise GCM or ESM output (further referred to as classical approach or method). For this purpose, we use the biome model BIOME1 (Prentice et al., 1992) that calculates the biome distribution in equilibrium to the input climate. As forcing, BIOME1 needs the monthly mean climatological precipitation, near-surface temperature and cloudiness, which were taken from each simulation considered in this study, respectively"

R: Section 2.2 It would be interesting to denote which simulations in Table 4 are run with dynamic vs static vegetation.

Our response: We mark the simulation with static vegetation.

R: Lines 154-156: Please could the authors briefly explain what is the difference between a gap model and a tiling approach. This will aid a reader who is less familiar with these models. Further, please could they describe how the PFT distribution has been

determined via the NPP of the vegetation categories for SEIB and why the vegetation categories do not already correspond to the PFT distribution?

Our response: We further explain the model SEIB. The calculation of the PFT fraction from the original (species-based) output was performed by the Japan Agency for Marine-Earth Science and Technology, Atmosphere and Ocean Research Institute (The University of Tokyo), and National Institute for Environmental Studies within the framework of CMIP5, the technique goes beyond the scope of this paper. We now write: "SEIB deviates from the other DGVMs in this study as it does not use the tiling approach of calculating PFT fractional coverage for each grid-cell. It is a so called forest gap-model, simulating the interactions among individual trees that compete for light and space in arising gaps (e.g. due to disturbances) within a spatially explicit virtual forest. The model was built for capturing the vegetation dynamics on local scale. The application of the model for larger (e.g. global) scales is possible, but global simulations partly disagreed with observations (Sato et al., 2007). The PFT distribution used in this study has been calculated in the post-processing for CMIP5 via the relative net primary productivity of the vegetation categories, it was not explicitly calculated by the model, which may lead to additional biases in the vegetation distribution."

R: Section 2.3 Line 175: Please could the authors briefly explain how Haxeltine and Prentice (1996) derived their vegetation compilation and what it includes?

Our response: This is not the scope of the paper, for details we refer to the publication of Haxeltine and Prentice (1996), Section 3.2.

R: Line 180: I do not understand this "on a basis of 5°C being higher than 900°C derived from modern observations". Please could the authors explain what they mean here? Does this correspond with Table 3? If so, it might be useful to say that. Our response: We further describe this: "RF99 additionally includes the biome 'Evergreen/Deciduous Mixed Forest/Woodland'. Here, this biome is assigned to the mega-biomes 'temperate forest' in warm regions and 'boreal forest' in colder regions

via the modern growing degree days distribution (GDD5≥900°C for warm region, GDD5<900°C for cold region, cf. Tab.2), derived from observations (University of East Anglia Climatic Research Unit Time Series 3.1, University of East Anglia, 2008, Harris et al., 2012)."

R: Line 203: "biomisation of simulated climate states (i.e. the classical method)". This is the first time the classical method has been referenced. It might be useful to the reader to describe it earlier and to detail how the authors' method differs from this classical method – particularly as it is used to compare the new method with throughout the results section. Otherwise, it appears somewhat out of the blue here.

Our response: We now mention the classical method already in the introduction (see comment above) and enlarge the part explaining the classical method in the Method part: "To assess the performance of the biomisation based on simulated PFTs, we additionally biomise the simulated climate fields corresponding to the PFT-distributions in each model. This is the conventionally used procedure to biomise GCM or ESM output (further referred to as classical approach or method). For this purpose, we use the biome model BIOME1 (Prentice et al., 1992) that calculates the biome distribution in equilibrium to the input climate. As forcing, BIOME1 needs the monthly mean climatological precipitation, near-surface temperature and cloudiness, which were taken from each simulation considered in this study, respectively. This classical biomisation approach can only handle climate data as input, the simulated PFT-distributions from the ESMs used in the here introduced PFT-based method are ignored. The original biomes has been grouped into the same mega-biome classification that is used for the PFT-based approach."

Results R: Section 3.1 It might be more constructive to compare the PFT- and classical climate- based biomisation methods for each model side by side in one figure, and not the former in Figure 3 and latter in Figure 4, given the authors compare the two. It is hard to compare each model given they are on separate pages.

Our response: We see this point. Our intention to plot all results based on the classical method on one page and the results based on the PFT-method on the other page was that we wanted to compare the methods and not the models. We did not change the order in the figures.

R: Line 287: It would be instructive to reference that "better represented" means in comparison to observations that are provided in Figure 2.

Our response: We now write: "As a consequence, the North American prairie fits better to observations for the PFT-based biome distributions."

Discussion R: Section 4.1 is well described and an honest and comprehensive view of the caveats of the methods.

Our response: Thank you.

Technical corrections R: Ni et al. (2010) missing most of the reference, Dallmeyer et al (2017) reference missing→check all references in the bibliography

Our response: We apologize that we had not update the reference list. We have now carefully checked all references in the text and add the missing references.

R: Line 97: additionally → additional Our response: done.

R: Line 147: has →have Our response: done.

R: Line 148: what is the importance of knowing that the simulations have been re-done on a new computer? Will they be better as a result?

Our response: We further explain it: The simulations used here has been conducted in a similar model-setup as described in Kleinen et. al. (2010), but has been re-done on a new computer (T. Kleinen, personal communication), which may lead to very small deviations from the original runs.

R: Line 159: extra → specifically Our response: done. R: Line 168: "than the other

simulations . . ." It would be helpful to add "that were run with dynamic/interactive vegetation". Our response: done. R: Line 187: details → detail Our response: done. R: Line 399: works →work Our response: done. R: Line 420: "very rudimentary" → "in a very rudimentary manner" Our response: done. R: Line 521: coverage→cover Our response: Do you mean L531? There we changed coverage to cover

---

## Author Comment (AC3) · 6 Nov 2018

R: I think that the paper of Dallmeyer et al. is a serious work and presents interesting findings in terms of method and results. The choice of Climate of the Past is, for me, appropriate: even if this paper mainly focus on vegetation models, it is also based on general circulation models. I think this paper is clear, well written and can be published in Climate of the Past with minor changes, which are listed below:

R: - abstract: must be reworked, it should be more informative on the methods, the results and more precise (method, key results, conclusions) : how many models are used, which is the principe of the method . . .

Our response: Thank you very much for the helpful comment, we revised the abstract

including more information on the method and its application in our study.

R: - introduction: this point concerns the originality of this paper, you need to better justified the objective and the questions of your paper given that at least two biomisation methods exist and work well (Prentice et al., 1992, 2011). The new biomisation method must be more clearly defined?

Our response: We agree with the Referee that the aim of providing a new method was not stressed enough. We now describe the problem with the classical method and the advantage of the new method more precisely: "... Using this classical method of biomisation, fundamental palaeo-vegetation analysis can be undertaken (e.g. Jolly et. al., 1998; Harrison et al., 2003; Wohlfahrt et al., 2008; Harrison et al., 2016, Dallmeyer et al., 2017) without requiring an explicit calculated vegetation distribution by the Earth System Models. On the other hand this also means that existing simulated plant functional type distributions calculated by the DGVMs being dynamically coupled in these models are neglected, since only the simulated climate pattern is taken into account. The biomisation via diagnostic biome models did not include any information on the original PFT-distribution simulated by the Earth System Models. As the DGVMs are generally more complex than the biome models and include more relevant processes, valuable information included in the PFT-distribution gets lost in the classical biomisiation by the biome models. A more appropriate way of biomisation would be to use directly the PFT-distributions calculated by the DGVMs." and: "To harmonize (palaeo)-vegetation distributions simulated by dynamic vegetation models and thereby facilitate the evaluation of Earth System Models (ESMs) and the comparison of model results and biome reconstructions, we developed a biomisation technique that is based on the PFT-distributions simulated by the DGVMs and few input variables and simple differentiation rules."

R: What is "natural" PFTS (line 36)?

Our response: we included '(non-anthropogenic)' as explanation for the natural.

R: Line 44, the ref Prentice et al., (1996) is needed.

Our response: This reference is included a line before.

Methods R: line 91: what is mean growing degree days? GDD0 or 5? Please define it.

Our response: We actually need both, GDD0 and GDD5. We add this.

R: Could you also define the "multi-year mean PFT cover fractions".

Our response: We further define it: "... multi-year mean PFT cover fractions (e.g. averaged over 100 years) are required."

R: Why don't you use the alpha parameter classically used in Prentice et al (1992, 1996).

Our response: The wetness is usually considered in the dynamic global vegetation models, i.e. in the cover fractions of the PFTs. In BIOME1 the alpha is needed e.g. to distinguish steppe and desert, but this differentiation is already done by the dynamic global vegetation models. A desert is a region in which only a small grid-cell fraction is covered by vegetation, etc. We only need the temperature limits for distinguishing the climatic zones (e.g. tropical vs. temperate vs. boreal).

R: Line 109: GDD0 exceeds 800°C or to the biome 'tundra', if GDD0 is below 800°C: could you explain the choice of these values?

Our response: we took these values from the BIOME4 model (i.e. the differentiation of steppe tundra and temperate grassland in BIOME4), we add the reference.

R: -simulations: line 121 "Simulations from nearly all state-of-the-art global dynamic vegetation models": the exact number of models is needed here as the name of the models.

Our response: We include the number and names of the models in this introductory sentence: Six different models could be considered (i.e. JSBACH, TRIFFID, OR-

CHIDEE, SEIB, LPJ and VECODE).

R: "Overall, eight simulations for the pre-industrial climate (PI) and vegetation, four for mid-Holocene (6k) conditions and five for Last Glacial Maximum (LGM) conditions have been used (Tab.4)": if I look at tab4, I find 6 simulations for PI, 3 for 6 ka, and 4 for LGM: please check; the format of tab 4 is not easy to read: correct it.

Our response: Thank you very much for this hint, this is indeed confusing. The original table includes the simulation names according to the references. We now change this column to 'period' to be less confusing and add the name of the simulations (where needed, i.e. Klockmann et al. Simulation) to the references. Please notice that Tab.4 is now Tab.3.

R: Line 140: what is CRUNCEP?

Our response: We further explain this: ...that is a combination of observations (CRU data) and reanalysis data (NCEP).

R: preparing the reference datasets: OK for the 6 ka and LGM but I have a problem with the comparison for the PI Period. The comparison with the Pre-industrial changes is an important point to validate the results. However, the models uses the pre-industrial period as a baseline whereas the pollen-inferred biomes use the late 20th century (modern pollen data used for the 0 ka in biome 6000 have been collected from 1960 to present day, so they don't correspond to preindustrial period). On the same way, in the RF99 dataset, Ramankutty and Foley used a global representation of permanent croplands in 1992, which not represent the PI period. This points of the discussion must be added and discussed in depth (CO2 bias and human impact).

Our response: We are aware of the temporal differences between the reference datasets and the model simulations and agree to the referee that this has to be clarified. We add the following information to the text: "Both vegetation datasets are derived for the modern time-slice not exactly corresponding to the pre-industrial period (around

1850 AD) simulated in the models. While the ice-sheet, the topography and the orbital conditions used for the pre-industrial control simulations are prescribed from modern conditions, greenhouse gases are set to pre-industrial values in the models. These differences in e.g. atmospheric CO2-concentration between the reference datasets and the simulations may lead to small discrepancies in the model. In addition, the references may be disturbed by anthropogenic influences."

R: -results: -lines 350-3522: you forget to discuss the increase of the temperate forest in Europe at 6 ka.

Our response: This is correct, but here we only discuss the pronounced changes. We did not change this.

discussion R: caveats in the method: line 401: the biome warm mixed forest is not only subtropical but also appears in Europe: please correct.

Our response: we now write: "warm-temperate forest (e.g. subtropical forest)"

R: Line 432: could you define the "anthropogenic plant functional types"?

Our response: we added "(i.e. land use)" to the text.

R: line 446: you state that the procedure of reconstructing paleovegetation often include modern analogue technique. I don't agree with that. The MAT is used to reconstruct the climate, not the biomes. The biome procedure using pollen follows the biomisation defined by prentice et al 1996, and peyron et al 1998 ...using a pollen-taxa –PFT assignment and a PFT-biome assignment which is done on modern samples (0 ka), and fossil samples (6 and LGM) independently. Please correct the text.

Our response: We agree that the term 'modern analogue' is inappropriate in this context. By this we meant that the transfer matrices from taxa into PFTs and from PFTs in Biomes were compiled on the basis of the recent vegetation. And these matrices do not have to be constant in time, i.e. they may not be applicable for glacial vegetation. Furthermore, the classification of the biomes was made on the basis of recent vegetation, but this classification does not have to correspond to the glacial classification, there were possibly other biomes that remain unconsidered. We changed the text to: Likewise, the matrices for the assignment of taxa into PFTs and from PFTs into Biomes have been constructed on the basis of the recent vegetation. These matrices do not have to be constant in time, i.e. they may not be applicable for glacial vegetation. Furthermore, the classification of the biomes itself corresponds to the modern vegetation and does not necessarily have to reflect the palaeo-vegetation. There might be other biomes in glacial climates that remain unconsidered.

R: biases in the PI ..., line 455: the classical method of biomising.. a ref is needed here.

Our response: We agree and now write: "The classical method of biomising climate states via biome models (here BIOME1 by Prentice et al., 1992)...“

R: Line 468: could you give more details on the sensitivity study performed by Dallmeyer et al 2017?

Our response: We agree to provide more details here. We now write: "A sensitivity study is performed following Dallmeyer et al. (2017) to relate differences in the biome distributions to precipitation or temperature deviations in the background climate (Fig.12). For this purpose, we successively replace the temperature or the precipitation in the CRU-TS4 forcing file for the BIOME1 model with the respective pre-industrial temperature or precipitation distributions simulated by the models. Afterwards, we compare the differences between the calculated biome distributions“

R: line 479: NPP ratios, not the acronym.

Our response: done.

R: Line 483: what do you mean by low score? Its not clear on the fig.

Our response: This is indeed misleading. We clarify this with: For the PFT-biomisation of CLIM-LPJ, k is basically reduced by an overestimation of boreal forest...

Figures R: too many figures! I strongly recommend to group the figures 2 and 3, and also the figures 9 and 11 ; the figures 4 and 8 can be moved to suppl. Material (not discussed in depth and not very useful).

Our response: We agree that we provide many figures, but we decided against grouping them so that we do not have to further reduce the size of the figures which are already quite small. Figure 4 is essential for the comparison between the classical and the new method, we keep both figures in the main text to facilitate the readability and comparability. Instead we move Fig 15 and the discussion of the comparison of the FPC-method and the PFT-method to the Appendix. We furthermore reduce the numbers of Tables.

References R: please check carefully your references, some are missing (Dallmeyer et al., 2017; Tian et al., 2017...) or not homogenized.

Our response: We homogenized the references and also add the missing references.

other points: R: in the text, you often write "pollen-based reconstructions" or "reconstructions"; I don't like this terminology because the pollen also is used to reconstruct climate or other : could you replace in the text by pollen-based (or inferred) biomes which is more precise and used.

Our response: We agree, this was not precise. We change the "reconstructions" to "pollen-based biome reconstruction" whenever it is relevant.

R: homogenize in the text: pi, PI, Pi -figures:

Our response: Thank you very much for carefully looking through the manuscript. We now use PI.

R: fig.1: Could you replace on the fig. "climate limits" by "bioclimatic limitation" and refer in the caption to tab.1 (climate limits) and tab 3 (Min cov. and Climate limits); the 4 simulated PFTs doesn't correspond to the text (p3: desert, forest, wood, grass and total vegetation): please check!

[Figure]

Our response: We adapt the figure according to the referees suggestion and change the PFT groups in the figure and the text and refer to Tab.2 including the detailed information.

R: fig.2 and others: could you change the color of grass? The green one is not very clear and can be confused with the temperate forest, orange will be better and commonly accepted by the biomisation community. This fig. must be group with fig.3.

Our response: We choose this colour set to be in line with our other publications. We do not change it. The Panel figures are quite small, to avoid a further reduction of the size we do not group Fig. 2 and 3.

R: fig.12: Not clear: what is the range of the climate factor? Could you explain better how they are calculated and not just refer to Dallmeyer et al. 2017?

Our response: The figure just shows, which climate input field leads to the differences in the biome distribution. Thus, the climate factors do not really have a range. We further explain the method in the text (see comment above) and also in the figure caption: "...The factors were calculated by performing a sensitivity test with the BIOME1 model following Dallmeyer et al. (2017) by successively replacing the temperature or the precipitation in the CRU-TS4.0 forcing file for BIOME1 with the respective data from the different PI simulations."

R: fig.15: even if it's explained in the text, its strange to see the biome savanna in Europe and Medit. Area (may be explain better in the fig. Caption)

Our response: We moved this comparison to the Appendix and also explain in the caption that the biome distribution inferred by Zhu includes temperate savanna. We write: "...Please notice that the FPC-method distinguishes temperate parkland, sclerophyll woodland and boreal parkland that all have been assigned in the mega-biome savanna, but savanna is only defined as tropical savanna in the PFT-method"

---

## Author Response (AR2)

**Response to the Referee's comments:**

We would like to thank the anonymous Referees for reviewing this manuscript again. Her/his constructive comments helped to further improve the manuscript.

**Reviewer:** This is my review of the revised version of this manuscript. Generally the authors have done a thorough job of responding to the reviewers' comments and the manuscript is substantially improved. I stand by my major comment on the previous version of this manuscript and am a little disappointed that the authors did not make an attempt to use Delta-V alongside Kappa to make a quantitative measure of the difference between vegetation maps and gridded maps and point-scale biomizations of pollen data. Even if the authors believe that the differences between mega-biomes are so substantial that Delta-V is not necessary, they effectively equate the difference between tropical rainforest and tundra as being qualitatively the same as that between temperate and boreal forests. At the very, very least, reference to Sykes et al. (1999) should be made in the manuscript, along with a call for using Delta-V in future studies (as the authors do in their responses to the reviewers document).

**Our response:** As noted in the previous response,  we consider the kappa statistic to be appropriate for the purposes of this study. Therefore, we have not re-evaluated the results using a different method. However, we agree with the reviewer that a comment on other evaluation metrics would be helpful. We have therefore added the following statement to the manuscript:

At this point it should be noted that we have selected the metrics in accordance with the research question of this study. For other purposes, such as estimating changes in biome distribution between present and future climate states, other metrics may be more appropriate, such as the Delta-V method, which also weights  changes in vegetation attributes (Sykes et al, 1999). The metrics used in this study do not differentiate how far the biomes deviate in their properties, e.g. differences between tropical forest and tundra are equated as being qualitatively the same as differences between temperate and boreal forest.

**R:** Nevertheless, the authors made a bona fide effort to respond to the rest of my comments and those of the other reviewer. While the manuscript is still rather long and tedious to read, and needs a thorough copyediting to improve the English language presentation, it should be acceptable for publication after minor revision. I have just two detailed comments on the revised manuscript below.

**Our response:** Climate of the Past will perform copy-editing by default.

**R:** Line 383-384

The statement "…warm-temperate forest shares most subtropical tree species with the tropical evergreen forest (Ni et al, 2010)" is simply not true, the reference to Ni et al. (2010) notwithstanding. Real warm-temperate forests are floristically very distinct from tropical evergreen forests, as I wrote in my previous comments. In the limited case of taxa that are commonly found in pollen spectra, there may be some commonality at the genera or family level, and in "model world" there may be overlap in the PFT definitions, but this statement is misleading and must be further qualified, or deleted.

**Our response:** We carefully discussed this statement again with experts for pollen-based biome reconstructions for China and further modified this sentences to:

The reconstructed biome 'warm-temperate forest' shares some subtropical PFTs with the tropical evergreen forest (Ni et al, 2010). These biomes are quite different in key species, but not on genus or family level, on which the pollen identification in the reconstructions is performed. Thus, these biomes tend to overlap in some regions and are sometimes mixed up in reconstructions (Chen et al, 2010).

**R:** Lines 528-533, 557-558, and Figure 6
It is interesting to note that the new PFT-based biomization method presented in this paper does not represent a large improvement over the traditional, offline climate-BIOME1/4-based method, except under certain circumstances such as the South American tropical forest area presented in Table D1. I suspect that in these places where there is a large difference between PFT-based and Climate-based may be because the vegetation models embedded in the ESMs are highly tuned to be adapted to the inherent biases in simulated climate present in the parent ESM, while an offline vegetation model is tuned to perform with the "observed" climate (e.g., from CRU). If the authors agree, it would be worth adding a sentence in the manuscript conclusions to acknowledge this point.
**Our response:** We agree with the Referee that DGVMs has been tuned, based on the climate simulated by ESMs, while the Biome-models such as BIOME1 or BIOME4 has been tuned to perform well with observed climate, so that DGVMs may better 'conceal' biases in the model, but we do not think, that this is relevant for this study. The PFT-based biomisation method and the Biome-models produce similar results because they use the same bioclimatic limits which define the biome belts quite well. The larger (more realistic) forest cover in the PFT-based biome distribution is related to the low limit of forest fraction needed in the assignment of the forest. We have already mentioned this in Section 4.3.
However, the impression seems to have arisen that we wanted to construct a better biomisation method than the classical method (via Biome-models). This was not our intention. We only wanted to develop a more direct method that could convert PFT distributions into biomes so that the additional information included due to the coupling with a DGVM would not get lost. And it is great, that this new method yield similar results as the classical, highly tuned method.
We further stress this by adding the following to the introduction (LL93):
The aim of developing this method was not to construct a better biomisation method than the classical method via diagnostic biome-models, but to develop a more direct method that can convert PFT distributions into biomes so that the additional information included due to the coupling of the DGVM will not get lost.